# CEP78 functions downstream of CEP350 to control biogenesis of primary cilia by negatively regulating CP110 levels

André Brás Gonçalves[1,‡], Sarah Kirstine Hasselbalch[1,†], Beinta Biskopstø Joensen[1,†], Sebastian Patzke[2], Pernille Martens[1], Signe Krogh Ohlsen[3], Mathieu Quinodoz[4,5,6], Konstantinos Nikopoulos[7], Reem Suleiman[1], Magnus Per Damsø Jeppesen[1], Catja Weiss[1], Søren Tvorup Christensen[1], Carlo Rivolta[4,5,6], Jens S Andersen[3], Pietro Farinelli[1*], Lotte Bang Pedersen[1*]

[1]Department of Biology, Section for Cell Biology and Physiology, University of Copenhagen, Copenhagen, Denmark; [2]Department of Radiation Biology, Institute for Cancer Research, Norwegian Radium Hospital, Oslo University Hospital, Oslo, Norway; [3]Department of Biochemistry and Molecular Biology, University of Southern Denmark, Odense, Denmark; [4]Institute of Molecular and Clinical Ophthalmology Basel (IOB), Basel, Switzerland; [5]Department of Ophthalmology, University of Basel, Basel, Switzerland; [6]Department of Genetics and Genome Biology, University of Leicester, Leicester, United Kingdom; [7]Department of Computational Biology, University of Lausanne, Lausanne, Switzerland

**\*For correspondence:**
Pietro.Farinelli@twelve.bio (PF);
lbpedersen@bio.ku.dk (LBP)

[†]These authors contributed equally to this work

**Present address:** [‡]Instituto de Medicina Molecular João Lobo Antunes, Faculdade de Medicina, Universidade de Lisboa, Lisbon, Portugal

**Competing interests:** The authors declare that no competing interests exist.

**Abstract** CEP78 is a centrosomal protein implicated in ciliogenesis and ciliary length control, and mutations in the *CEP78* gene cause retinal cone-rod dystrophy associated with hearing loss. However, the mechanism by which CEP78 affects cilia formation is unknown. Based on a recently discovered disease-causing *CEP78* p.L150S mutation, we identified the disease-relevant interactome of CEP78. We confirmed that CEP78 interacts with the EDD1-DYRK2-DDB1[VPRBP] E3 ubiquitin ligase complex, which is involved in CP110 ubiquitination and degradation, and identified a novel interaction between CEP78 and CEP350 that is weakened by the CEP78[L150S] mutation. We show that CEP350 promotes centrosomal recruitment and stability of CEP78, which in turn leads to centrosomal recruitment of EDD1. Consistently, cells lacking CEP78 display significantly increased cellular and centrosomal levels of CP110, and depletion of CP110 in CEP78-deficient cells restored ciliation frequency to normal. We propose that CEP78 functions downstream of CEP350 to promote ciliogenesis by negatively regulating CP110 levels via an EDD1-dependent mechanism.

## Introduction

Primary cilia are antenna-like sensory organelles that play pivotal roles in coordinating various signaling pathways important for human development and tissue homeostasis (*Anvarian et al., 2019*). They comprise a microtubule-based axoneme core, which extends directly from the mother centriole-derived basal body and is surrounded by a bilayer membrane enriched for specific receptors, ion channels, and lipids that endow the organelle with unique signaling properties (*Nachury and Mick, 2019*). Not surprisingly, mutations in genes that affect assembly, structure, or function of cilia are causative for a growing number of pleiotropic diseases and syndromes called ciliopathies, which include cone-rod dystrophy in the retina and hearing loss (CRDHL; MIM# 617236) amongst others (*Reiter and Leroux, 2017*). Ciliopathy genes include those coding for components of the

centrosome, which contains the daughter and mother centriole and gives rise to the ciliary basal body. The mother centriole is distinguished from the daughter centriole by distal and subdistal appendages, which play critical roles in vesicle docking at the onset of ciliogenesis and in microtubule anchoring, respectively. In addition, the centrosome contains pericentriolar material and is associated with centriolar satellites that affect cilia biogenesis and function in various ways (*Breslow and Holland, 2019*).

Assembly of primary cilia is a complex, multistep process that is tightly coordinated with the cell cycle. In actively proliferating cells, centriolar coiled coil protein 110 (CP110; also known as CCP110) and centrosomal protein of 97 kDa (CEP97) cap the distal ends of both mother and daughter centrioles and suppress ciliogenesis. Furthermore, overexpression of CP110 in growth-arrested cells prevents ciliogenesis (*Spektor et al., 2007*). CP110 also regulates centriole duplication and length control during S phase and interacts with key regulators of centriole duplication, including PLK4 (*Chen et al., 2002*; *Kleylein-Sohn et al., 2007*; *D'Angiolella et al., 2010*; *Li et al., 2013*). As cells enter G1/G0, ciliogenesis begins with recruitment of pre-ciliary vesicles to the distal end of the mother centriole. The vesicles subsequently fuse to form a larger ciliary vesicle underneath which the ciliary transition zone and axoneme are assembled. The axoneme is further extended by intraflagellar transport (IFT), and the ciliary vesicle expands and matures to form the ciliary membrane and a surrounding sheath that fuses with the plasma membrane upon completion of ciliogenesis (*Sorokin, 1962*; *Sorokin, 1968*; *Shakya and Westlake, 2021*).

Initiation of transition zone formation and axoneme extension during early stages of ciliogenesis require removal of the CEP97-CP110 complex from the distal end of the mother centriole (*Spektor et al., 2007*). This process relies on M-phase phosphoprotein 9 (MPP9), which interacts directly with CEP97 and cooperates with kinesin KIF24 to recruit the CEP97-CP110 complex to the distal centriole end (*Kobayashi et al., 2011*; *Huang et al., 2018*). During ciliogenesis, phosphorylation of MPP9 by Tau tubulin kinase 2 (TTBK2) leads to degradation of MPP9 by the ubiquitin proteasome system (UPS), which results in destabilization of the CEP97-CP110 complex causing its removal from the distal end of the mother centriole (*Huang et al., 2018*). TTBK2 is recruited to the mother centriole distal appendages by CEP164 (*Čajánek and Nigg, 2014*; *Oda et al., 2014*), where it also phosphorylates CEP83 to promote CP110 removal (*Lo et al., 2019*). Consequently, depletion of CEP164, TTBK2, CEP83, or other centriole distal appendage proteins that regulate their localization and/or function impairs ciliogenesis (*Graser et al., 2007*; *Goetz et al., 2012*; *Schmidt et al., 2012*; *Tanos et al., 2013*; *Ye et al., 2014*; *Kurtulmus et al., 2018*). Mother centriole recruitment of TTBK2 and removal of the distal CEP97-CP110 cap additionally require the subdistal CEP350-FOP-CEP19 complex, which furthermore interacts with RABL2 and IFT-B complex components to promote their axonemal entry (*Kanie et al., 2017*; *Nishijima et al., 2017*; *Mojarad et al., 2017*).

Despite recent advances, the precise mechanisms by which CP110 regulates ciliogenesis and is removed from the mother centriole at the onset of ciliogenesis remain incompletely understood. For example, a recent study implicated the homologous to the E6AP carboxyl terminus (HECT)-type EDD1-DYRK2-DDB1^VPRBP E3 ligase complex in ubiquitination and degradation of CP110 via a mechanism involving direct interaction of viral protein R binding protein (VPRBP; also known as DCAF1) and centrosomal protein of 78 kDa (CEP78) (*Hossain et al., 2017*). Specifically, the authors reported that CEP78 suppresses CP110 ubiquitination by EDD1 (also known as UBR5 and EDD), and it was proposed that CP110 is phosphorylated by DYRK2 and thereby recognized and brought close to EDD1 by VPRBP. EDD1 then transfers ubiquitin to CP110, leading to its degradation. When CEP78 binds VPRBP, it induces a conformational change in the complex, thereby preventing CP110 ubiquitination. Further, they demonstrated that depletion of CEP78 promoted centriole elongation, whereas *CEP78* overexpression inhibited primary cilia formation in hTERT-immortalized retinal pigment epithelial (RPE1) cells (*Hossain et al., 2017*). On the other hand, knockout (KO) of *Dyrk2* in the mouse was reported to result in elongation of primary cilia, but centrosomal CP110 levels appeared unaffected in *Dyrk2* mouse KO cells (*Yoshida et al., 2020*). Therefore, it remains uncertain how CEP78 and the EDD1-DYRK2-DDB1^VPRBP complex affect CP110 homeostasis to control ciliogenesis.

*CEP78* is composed of 16 exons and encodes a protein comprising 722 amino acids that possesses five consecutive leucine-rich repeats (LRR) at the N-terminus and a coiled-coil (CC) domain at the C-terminus (*Nikopoulos et al., 2016*; *Ascari et al., 2020*). Four independent studies have reported eight different mutations in *CEP78* in patients with CRDHL (*Nikopoulos et al., 2016*; *Ascari et al., 2020*; *Namburi et al., 2016*; *Fu et al., 2017*), whereas one study identified a

homozygous *CEP78*-truncating variant in a family with non-syndromic retinitis pigmentosa (MIM# 268003; *de Castro-Miró et al., 2016*), another form of retinal degeneration. Of these studies, two have investigated the functional consequences of human *CEP78* mutations at the cellular level. In one study, whole-exome sequencing (WES) identified biallelic mutations in *CEP78* in two unrelated families from Greece and Sweden, respectively (*Nikopoulos et al., 2016*). The Greek subject had a homozygous base substitution at the splice donor site in intron 3 of *CEP78* (NM_032171.2:c.499 +1G>T). Two subjects from a Swedish family carried heterozygous mutations, one base substitution in intron 3 (NM_032171.2:c.499+5G>A) and a single-nucleotide deletion in exon 5 (NM_032171.2: c.633del; p.Trp212GlyfsTer18) causing a frameshift. These *CEP78* mutations lead to exon skipping and premature stop codons accompanied by almost undetectable levels of CEP78 protein in human skin fibroblasts (HSFs) of affected individuals. Furthermore, it was found that HSFs from these patients have significantly longer primary cilia as compared to control cells (*Nikopoulos et al., 2016*). More recently, a missense mutation in *CEP78* (NM_032171.2:c.449T>C; p.Leu150Ser) was identified in three unrelated families from Belgium and Germany diagnosed with CRDHL (*Ascari et al., 2020*). In the two Belgian families, affected individuals were homozygous for the p. Leu150Ser variant, whereas affected individuals from the German family displayed compound heterozygosity for this variant and a novel splice site variant, NM_032171.2:c.1462–1G>T. In all cases, HSFs from patients harboring the p.Leu150Ser mutation (hereafter: L150S) displayed decreased cellular levels of CEP78 and significantly elongated cilia compared to control HSFs (*Ascari et al., 2020*), as seen in patient-derived HSFs with CEP78 truncating mutations (*Nikopoulos et al., 2016*). In addition, other ciliopathy features were reported in some of the affected individuals with the CEP78[L150S] mutation, including obesity, respiratory problems, diabetes 2, and infertility (*Ascari et al., 2020*). In summary, available data derived from human patients indicates that depletion of CEP78 leads to elongation of primary cilia at the cellular level, which manifests in ciliopathy phenotypes at the organism level. However, the mechanism by which CEP78 regulates ciliary length is not known.

In addition to the patient studies described above, some studies have addressed CEP78 function in different cell culture models. First, a study showed that siRNA-mediated depletion of CEP78 in RPE1 cells reduces the frequency of cells forming primary cilia, possibly due to centriolar anchoring defects, but the length of the residual cilia was not assessed. Similarly, depletion of CEP78 in *Planarians* was shown to impair motile cilia formation due to defective docking of centrioles to the cell surface (*Azimzadeh et al., 2012*). Another study identified CEP78 interaction with PLK4 and implicated CEP78 in PLK4-mediated centriole over duplication, whereas possible roles for CEP78 in relation to cilia were not addressed (*Brunk et al., 2016*). Finally, as mentioned above, a study reported that CEP78 directly interacts with VPRBP of the EDD1-DYRK2-DDB1[VPRBP] E3 ligase complex to suppress CP110 ubiquitination and thereby stabilize it (*Hossain et al., 2017*). How such CEP78-mediated stabilization of CP110 might lead to the long cilia phenotype observed in patient fibroblasts lacking CEP78 (*Nikopoulos et al., 2016*; *Ascari et al., 2020*) and/or the reduced ciliation frequencies seen in CEP78-depleted *Planarians* and RPE1 cells (*Azimzadeh et al., 2012*) is unclear.

Here, we show that in RPE1 cells and patient-derived HSFs loss of CEP78 leads to reduced ciliation frequency as well as increased length of the cilia that do form. Further, we find that a mutant line expressing a partially functional, truncated version of CEP78 displays reduced ciliation frequency but normal length of residual cilia. By taking advantage of the recently identified disease-causing CEP78[L150S] mutation (*Ascari et al., 2020*), we used a quantitative mass spectrometry-based approach to identify a disease-relevant interactome of CEP78. We confirmed that CEP78 interacts with the EDD1-DYRK2-DDB1[VPRBP] complex implicated in CP110 ubiquitination and degradation (*Hossain et al., 2017*), and furthermore identified a novel interaction between CEP78 and the N-terminal region of CEP350. The interaction of CEP78 with both VPRBP and CEP350, as well as centrosomal recruitment of CEP78, is dramatically reduced by the CEP78[L150S] mutation. Lack of centrosomal recruitment of CEP78[L150S] is likely due to impaired interaction with CEP350 since centrosomal and cellular levels of CEP78 were significantly decreased in *CEP350* KO cells. Conversely, cells lacking CEP78 displayed significantly decreased centrosomal levels of EDD1 and unaltered or slightly increased centrosomal levels of VPRBP and CEP350. In addition, CEP78-deficient cells showed significantly increased cellular and centrosomal levels of CP110, presumably owing to the reduced centrosomal levels of EDD1 observed in these cells. Depletion of CP110 in CEP78-deficient cells restored ciliation frequency, but not the increased length of remaining cilia, to normal. Collectively our results suggest that CEP78 functions downstream of CEP350 to promote cilia biogenesis

by negatively regulating CP110 levels via an EDD1-dependent mechanism, but suppresses ciliary elongation independently of CP110.

## Results

### Depletion of CEP78 leads to fewer and longer primary cilia in cultured cells

Previous studies showed that patient-derived fibroblasts lacking CEP78 display significantly elongated primary cilia compared to control cells (*Nikopoulos et al., 2016*; *Ascari et al., 2020*), whereas depletion of CEP78 in *Planarians* and RPE1 cells was shown to significantly reduce cilia numbers (*Azimzadeh et al., 2012*). To reconcile these seemingly contradictory findings, we analyzed cilia numbers and length in serum-deprived wildtype (WT) RPE1 cells and four different *CEP78* KO clones generated by CRISPR/Cas9 methodology. We first confirmed by western blot analysis with CEP78-specific antibody that endogenous CEP78 is lacking in three of the mutant clones, designated #2, #52, and #73, whereas clone #44 expresses a shorter version of CEP78 (*Figure 1—figure supplement 1A*). Sequencing of clone #44 indicated that it contains an insertion of the px459-Cas9 plasmid in exon 1 of *CEP78*, suggesting that a shorter version of CEP78 lacking the extreme N-terminus is expressed in this strain, possibly from an alternative promoter. Immunofluorescence microscopy (IFM) analysis with antibodies against ciliary (ARL13B) and centrosomal (CEP350) markers showed that the frequency of ciliated cells is significantly reduced in all four *CEP78* KO clones compared to WT cells (*Figure 1A, B*; *Figure 1—figure supplement 1B*). In addition, the length of the remaining cilia was significantly increased in the three *CEP78* KO clones that completely lack CEP78, but identical to the average WT cilia size in clone #44 (*Figure 1A, C*; *Figure 1—figure supplement 1C*). The latter result indicates that the truncated CEP78 protein expressed in clone #44 functions normally with respect to ciliary length control, in turn suggesting that CEP78 may affect ciliogenesis and ciliary length by distinct mechanisms. For the rest of this article, RPE1 *CEP78* KO cells refer to clone #73 unless otherwise indicated.

In agreement with our observations in RPE1 cells, similar results were obtained using previously described patient-derived *CEP78*-deficient or control HSFs (*Figure 1D–F*; *Nikopoulos et al., 2016*; *Ascari et al., 2020*). The low ciliation frequencies of serum-deprived *CEP78* mutant cells were not secondary to cell cycle defects, as judged by western blot analysis with antibody against retinoblastoma protein phosphorylated at S807/811 (P-Rb; *Figure 1G–L*). However, in serum-fed cells, P-Rb levels were significantly higher in *CEP78* mutant cells compared to controls (*Figure 1G–L*), indicating that mutant cells may progress slower through S-phase (*Knudsen and Wang, 1997*). This is in line with previous reports indicating that CEP78 protein levels are upregulated in late S-G2 phase, suggesting a role for CEP78 at this cell cycle stage, for example, during centriole duplication (*Hossain et al., 2017*; *Brunk et al., 2016*). We conclude that CEP78-deficient RPE1 and HSF cells display reduced ciliation frequencies as well as increased length of remaining cilia, thus reconciling previous observations (*Nikopoulos et al., 2016*; *Ascari et al., 2020*; *Azimzadeh et al., 2012*).

### Analysis of RPE1 WT and *CEP78* KO cells expressing FLAG- or mNG-tagged CEP78 or CEP78$^{L150S}$ fusions

To confirm the above results, we first set out to perform a rescue experiment by expressing FLAG-CEP78 in WT and *CEP78* KO RPE1 cells followed by serum-deprivation and IFM analysis using FLAG and ARL13B antibodies. In parallel, we performed similar experiments with FLAG-CEP78$^{L150S}$. We first analyzed the localization and expression levels of the two FLAG fusions in transiently transfected serum-deprived RPE1 cells. In 4% paraformaldehyde (PFA)-fixed cells, we observed that the ARL13B antibody labeled the cilium itself as well as the basal body, but not the daughter centriole. The basal body pool of ARL13B likely corresponds to that present in the mother centriole-associated vesicle of mitotic centrosomes (*Paridaen et al., 2013*). The transiently expressed FLAG-CEP78 fusion protein localized to both centrioles at the base of cilia as expected (*Hossain et al., 2017*; *Nikopoulos et al., 2016*; *Brunk et al., 2016*), whereas centriolar localization of FLAG-CEP78$^{L150S}$ was severely compromised although not completely abolished (*Figure 2A, B*). The latter result is in line with previous reports showing that the LRR region in the N-terminus of CEP78, which encompasses residue L150, is important for its localization to centrioles (*Hossain et al., 2017*; *Brunk et al., 2016*). Both FLAG

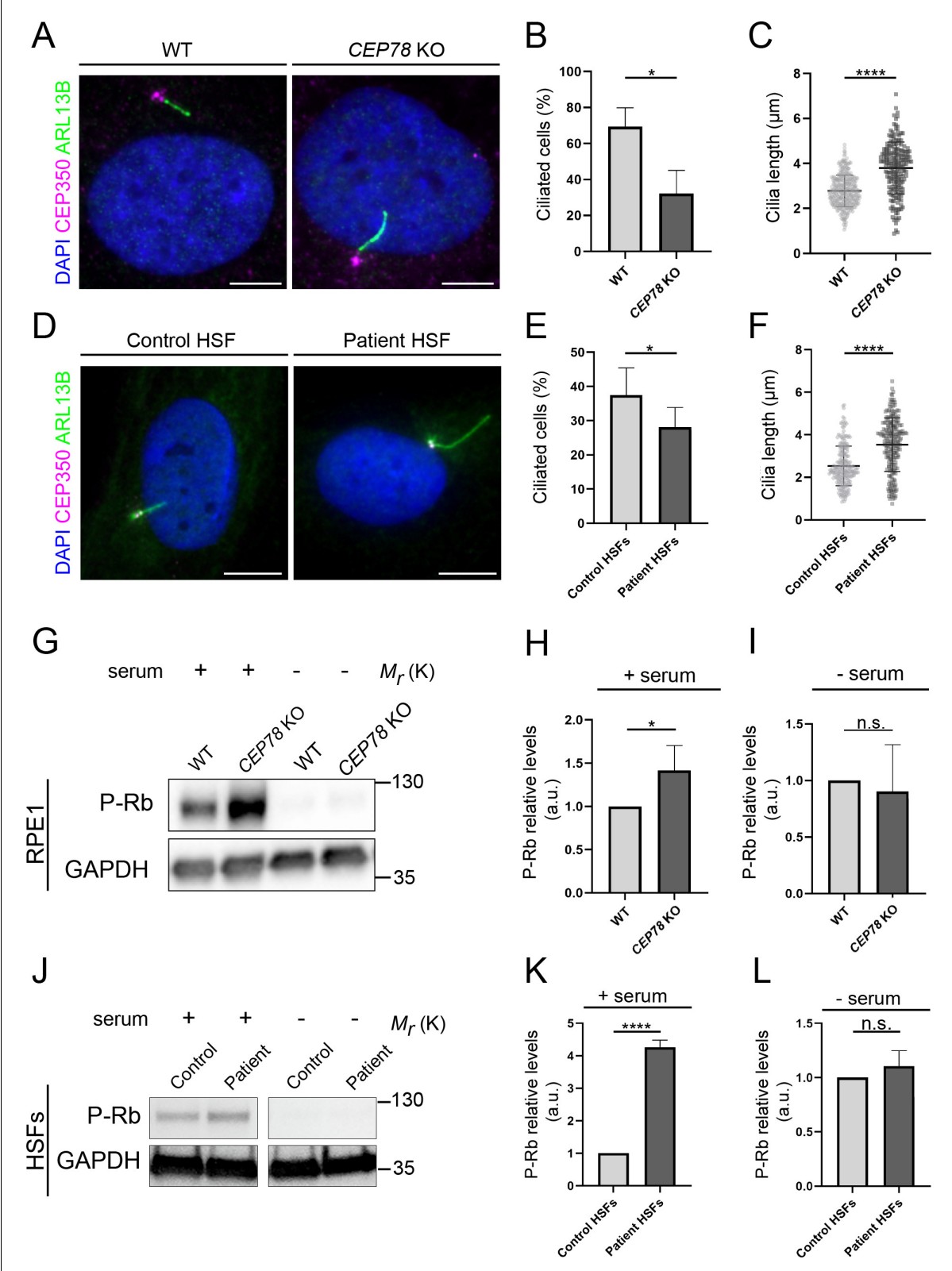

**Figure 1.** CEP78 mutant cells display fewer, but longer primary cilia. (A) Representative immunofluorescence microscopy (IFM) images of serum-deprived RPE1 wildtype (WT) and *CEP78* knockout (KO) cells stained with the indicated antibodies; DAPI marks the nucleus. Scale bar, 5 µm. (B, C) Quantification of the percentage of ciliated cells (B) and the length of residual cilia (C) in WT and *CEP78* KO cells, based on images as shown in (A). Data were normalized in relation to WT values. Student's t-test (two-tailed, unpaired) was used for the statistical analysis. Graphs in (B) represent

*Figure 1 continued on next page*

*Figure 1 continued*

accumulated data from three individual experiments (n = 384 for WT and n = 322 for *CEP78* KO cells). Graphs in (C) show data from three individual experiments (n = 338 for WT and n = 198 for *CEP78* KO cells). (D) Representative IFM images of serum-deprived control and *CEP78* mutant (Patient) human skin fibroblasts (HSFs) stained with the indicated antibodies; DAPI marks the nucleus. Scale bar, 5 μm. (E, F) Quantification of the percentage of ciliated cells (E) and the length of residual cilia (F) in control and *CEP78* patient HSFs (data from HSFs derived from patient 2702 r34, individual II-3; *Nikopoulos et al., 2016*), based on images as shown in (D). Graphs in (E) represent accumulated data from seven individual experiments (n = 678 for control HSFs; n = 707 for patient-derived HSFs). Graphs in (F) show data from seven individual experiments (n = 237 for control HSFs; n = 216 for patient-derived HSFs). Data were normalized in relation to control values. A Student's t-test (unpaired, two-tailed) was performed to address differences in the ciliary frequency and length between control and patient HSFs. (G) Western blot analysis of Rb phosphorylated on S807/811 (P–Rb) in RPE1 WT and *CEP78* KO cells grown with or without serum. (H, I) Quantification of data shown in (G) from three independent experiments analyzed in duplicates. (J) P-Rb blots from HSFs derived from control and patient HSFs grown with or without serum (data from HSFs derived from patient F3: II:1; *Ascari et al., 2020*). GAPDH was used as a loading control. (K, L) Quantification of data shown in (J) from three independent experiments analyzed in duplicates. Student's t-test (two-tailed, unpaired) was used for the statistical analysis in (H, I) and (K, L). Error bars of graphs represent SD and data are shown as mean ± SD. a.u., arbitrary units; *p<0.05; ****p<0.0001; n.s., not statistically significant.

The online version of this article includes the following source data and figure supplement(s) for figure 1:

**Source data 1.** Original western blots corresponding to *Figure 1G*.
**Source data 2.** Original western blots corresponding to *Figure 1J*.
**Figure supplement 1.** Western blot analysis and ciliary frequency and length in RPE1 wildtype (WT) and *CEP78* knockout (KO) clones.
**Figure supplement 1—source data 1.** Original western blots for *Figure 1—figure supplement 1*.

fusions were expressed at similar levels in the cells (*Figure 2—figure supplement 1*); this was somewhat surprising given that endogenous CEP78$^{L150S}$ is largely undetectable by western blot analysis of patient-derived HSFs, suggesting its stability is compromised (*Ascari et al., 2020*). Therefore, we tested if the L150S mutation might affect binding of the CEP78 antibody used in *Ascari et al., 2020* to CEP78. However, FLAG immunoprecipitation (IP) and western blot analysis of 293T cells expressing FLAG-CEP78 or FLAG-CEP78$^{L150S}$ indicated that the CEP78 antibody binds equally well to WT CEP78 and the CEP78$^{L150S}$ mutant protein. This analysis also revealed that endogenous CEP78 co-IPs with FLAG-CEP78, indicating that CEP78 can form homodimers/oligomers (*Figure 2—figure supplement 2*). These results indicate that the L150S mutation compromises the recruitment of CEP78 to the centrosome, but not its short-term stability, at least in RPE1 cells. We cannot rule out that the N-terminal FLAG tag may stabilize FLAG-CEP78$^{L150S}$ and prevent it from being degraded.

Next, we tested if FLAG-CEP78 or FLAG-CEP78$^{L150S}$ could rescue the ciliary phenotypes of *CEP78* KO cells. However, upon transient expression of the FLAG-CEP78 fusions we could not rescue the cilia frequency and length phenotype of the *CEP78* KO cells, and cilia frequency and length were also affected in WT control cells transiently expressing these fusions (data not shown). We therefore generated cell lines stably expressing mNeonGreen (mNG)-tagged CEP78 or CEP78$^{L150S}$ at ca. 3–4 times the endogenous CEP78 level by viral transduction of mNG-CEP78 constructs into WT and *CEP78* KO RPE1 cells, respectively (*Figure 2—figure supplement 3A*). Microscopic examination of these lines indicated that mNG-CEP78 was concentrated at the centrosome as expected, whereas mNG-CEP78$^{L150S}$ was diffusely located in the cytosol (*Figure 2—figure supplement 3B*), in agreement with results for transiently expressed FLAG-CEP78/CEP78$^{L150S}$ fusions (*Figure 2A, B*). Importantly, stably expressed mNG-CEP78 could fully rescue the ciliation frequency phenotype of *CEP78* KO cells, whereas mNG-CEP78$^{L150S}$ could not (*Figure 2C*). In addition, we observed that WT cells stably expressing mNG-CEP78 have significantly reduced ciliation frequency compared to untransfected WT cells (*Figure 2C*), indicating that mild overexpression of mNG-CEP78 seems to inhibit ciliogenesis. Furthermore, WT or *CEP78* KO cells stably expressing mNG-CEP78 had significantly shorter cilia than untransfected WT and *CEP78* KO cells, or *CEP78* KO cells expressing mNG-CEP78$^{L150S}$. However, we note that cilia length in the latter was significantly shorter than untransfected *CEP78* KO cells (*Figure 2D*). Together, these results suggest that mNG-CEP78 is able to fully rescue the cilia frequency phenotype of *CEP78* KO cells, whereas mNG-CEP78$^{L150S}$ is not. Furthermore, both mNG-CEP78 and mNG-CEP78 $^{L150S}$ promote ciliary shortening, but mNG-CEP78 does it more efficiently than mNG-CEP78 $^{L150S}$. We conclude that the observed ciliary frequency and length phenotypes of the *CEP78* KO cells are caused specifically by CEP78 loss, and that the L150S mutations impairs CEP78 centrosome localization and function.

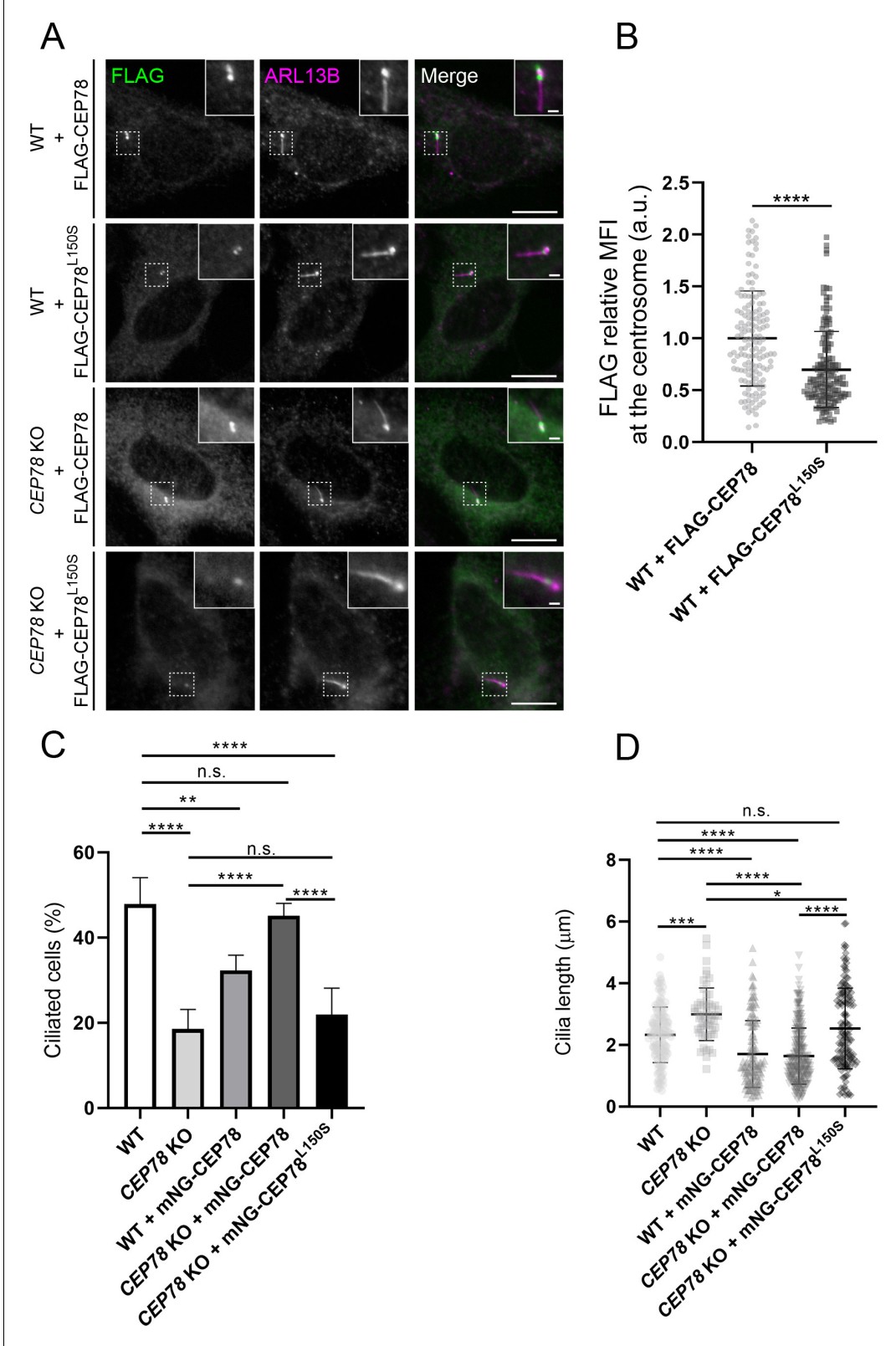

**Figure 2.** Analysis of RPE1 wildtype (WT) and *CEP78* knockout (KO) cells expressing FLAG- or mNG-tagged CEP78 or CEP78$^{L150S}$ fusions. (**A**) Representative immunofluorescence microscopy (IFM) images of serum-starved WT and *CEP78* KO cells transiently expressing FLAG-CEP78 and FLAG-CEP78$^{L150S}$ labeled with antibodies against FLAG (green) and ARL13B (magenta). Insets show enlarged views of the cilia-centrosome region. Scale bars: 5 µm in original images and 1 µm in magnified images. (**B**) Quantification of the relative mean fluorescence intensity (MFI) of FLAG-CEP78 and FLAG-

*Figure 2 continued on next page*

*Figure 2 continued*

CEP78$^{L150S}$ at the centrosome in serum-starved WT cells. Accumulated data from three individual experiments (n = 145 and n = 151 for cells transfected with FLAG-CEP78 and FLAG-CEP78$^{L150S}$, respectively). Student's t-test (two-tailed, unpaired) was used for the statistical analysis. Data is shown as mean ± SD. (C, D) Quantification of the percentage of ciliated cells (C) or ciliary length (D) in serum-deprived WT and *CEP78* KO cells stably expressing mNG-CEP78 or mNG-CEP78$^{L150S}$. Analysis on cilia numbers and length was performed through accumulated data from three independent experiments for WT, *CEP78* KO, and WT + mNG-CEP78 cells and five independent experiments for *CEP78* KO cells expressing either the mNG-CEP78 or the mNG-CEP78$^{L150S}$ (for ciliary frequency: n = 304 for WT; n = 322 for *CEP78* KO; n = 425 for WT + mNG-CEP78; n = 621 for C*EP78* KO + mNG-CEP78, and n = 638 for C*EP78* KO + mNG-CEP78$^{L150S}$; for ciliary length: n = 145 for WT; n = 56 for *CEP78* KO; n = 138 for WT + mNG-CEP78; n = 280 for C*EP78* KO + mNG-CEP78 and n = 141 for C*EP78* KO + mNG-CEP78$^{L150S}$). Ordinary one-way ANOVA with Dunnett's multiple comparison test was used for the statistical analysis of the ciliary frequency and length amongst all groups, in relation to the mean of WT cells, designated as the control group. Differences between *CEP78* KO and *CEP78* KO+ mNG-CEP78; *CEP78* KO and *CEP78* KO + mNG-CEP78$^{L150S}$ and *CEP78* KO + mNG-CEP78 and *CEP78* KO + mNG-CEP78$^{L150S}$ were addressed in a pairwise fashion using an unpaired and two-tailed Student's t-test. Error bars in (C) indicate SD and data in (D) is presented as mean ± SD. *$p<0.05$; **$p<0.01$; ***$p<0.001$; ****$p<0.0001$; n.s., not statistically significant; a.u., arbitrary units.

The online version of this article includes the following source data and figure supplement(s) for figure 2:

**Figure supplement 1.** Expression of FLAG-CEP78 and FLAG-CEP78$^{L150S}$ in RPE1 wildtype (WT) and *CEP78* knockout (KO) cells.

**Figure supplement 1—source data 1.** Original western blots for *Figure 2—figure supplement 1*.

**Figure supplement 2.** CEP78 endogenous antibody binds equally well to FLAG-CEP78 WT and L150S mutant fusions.

**Figure supplement 2—source data 1.** Original western blots for *Figure 2—figure supplement 2*.

**Figure supplement 3.** Stable expression of mNG-CEP78 and mNG-CEP78$^{L150S}$ in RPE1 wildtype (WT) and *CEP78* knockout (KO) cells.

**Figure supplement 3—source data 1.** Original western blots for *Figure 2—figure supplement 3*.

## CEP78 interacts with the EDD1-DYRK2-DDB1$^{VPRBP}$ complex and CEP350

To explore how CEP78 is recruited to centrioles to regulate ciliary frequency and length, we first used Stable Isotope Labeling by Amino acids in Cell culture (SILAC)-based quantitative mass spectrometry (MS) to identify the differential interactomes of FLAG-CEP78 and FLAG-CEP78$^{L150S}$ expressed in 293T cells (see Materials and methods for details). We identified interaction between CEP78 and several components of the EDD1-DYRK2-DDB1$^{VPRBP}$ complex (*Figure 3A*), in agreement with previous work showing direct binding of CEP78 to VPRBP within this complex (*Hossain et al., 2017*). Interestingly, the MS results indicated that the CEP78$^{L150S}$ mutation dramatically reduces this interaction (*Figure 3A*), which we confirmed by IP and western blot analysis of 293T cells co-expressing Myc-VPRBP and FLAG-CEP78 or FLAG-CEP78$^{L150S}$ (*Figure 3B*). In addition, our interactome analysis identified CEP350 as potential novel CEP78 binding partner (*Figure 3A*). Using IP and western blot analysis in 293T cells co-expressing FLAG-CEP78 or FLAG-CEP78$^{L150S}$ with a Myc-tagged N-terminal or C-terminal CEP350 truncation, we found that CEP78 binds to the N-terminal region of CEP350 (residues 1–983; *Figure 3C, E*); reciprocal co-IP with Myc-tagged CEP350 constructs confirmed this result (*Figure 3—figure supplement 1A*). Furthermore, we found that the interaction between CEP78 and CEP350 N-terminus is reduced by the CEP78$^{L150S}$ mutation (*Figure 3C*). Using a similar approach, we could not detect interaction between CEP78 and endogenous FOP (*Figure 3C*), whereas under similar conditions endogenous FOP was readily co-IPed with the C-terminus of CEP350 (*Figure 3—figure supplement 1A*), in agreement with a previous report (*Yan et al., 2006*). We also mapped the CEP350 interaction site in CEP78 and found that CEP350 primarily binds to the C-terminal region of CEP78 comprising residues 395–722 (*Figure 3D, E*), although a weak interaction with a fragment comprising the entire LRR region (residues 2–403) was also observed. In contrast, truncation of the latter fragment into two smaller fragments abolished binding to the CEP350 N-terminus (*Figure 3D, E*). Binding of CEP78 to VPRBP similarly requires the entire LRR region of CEP78 (*Hossain et al., 2017*), and we therefore wondered whether interaction between CEP350 and CEP78 might be mediated by VPRBP. However, Myc IP experiments and western blot analysis failed to reveal physical interaction between VPRBP and Myc-CEP350 N-terminus (*Figure 3—figure supplement 1A*). Moreover, GFP IP of cells co-expressing GFP-CEP78 and Myc-CEP350 N- or C-terminal fragments showed that GFP-CEP78 can bind to Myc-CEP350 N-terminus and endogenous VPRBP at the same time, indicating that CEP350 does not compete with VPRBP for binding to CEP78 (*Figure 3—figure supplement 1B*). We conclude that CEP78 binds independently to VPRBP and the N-terminus of CEP350 and that both of these interactions are reduced by the CEP78$^{L150S}$ mutation.

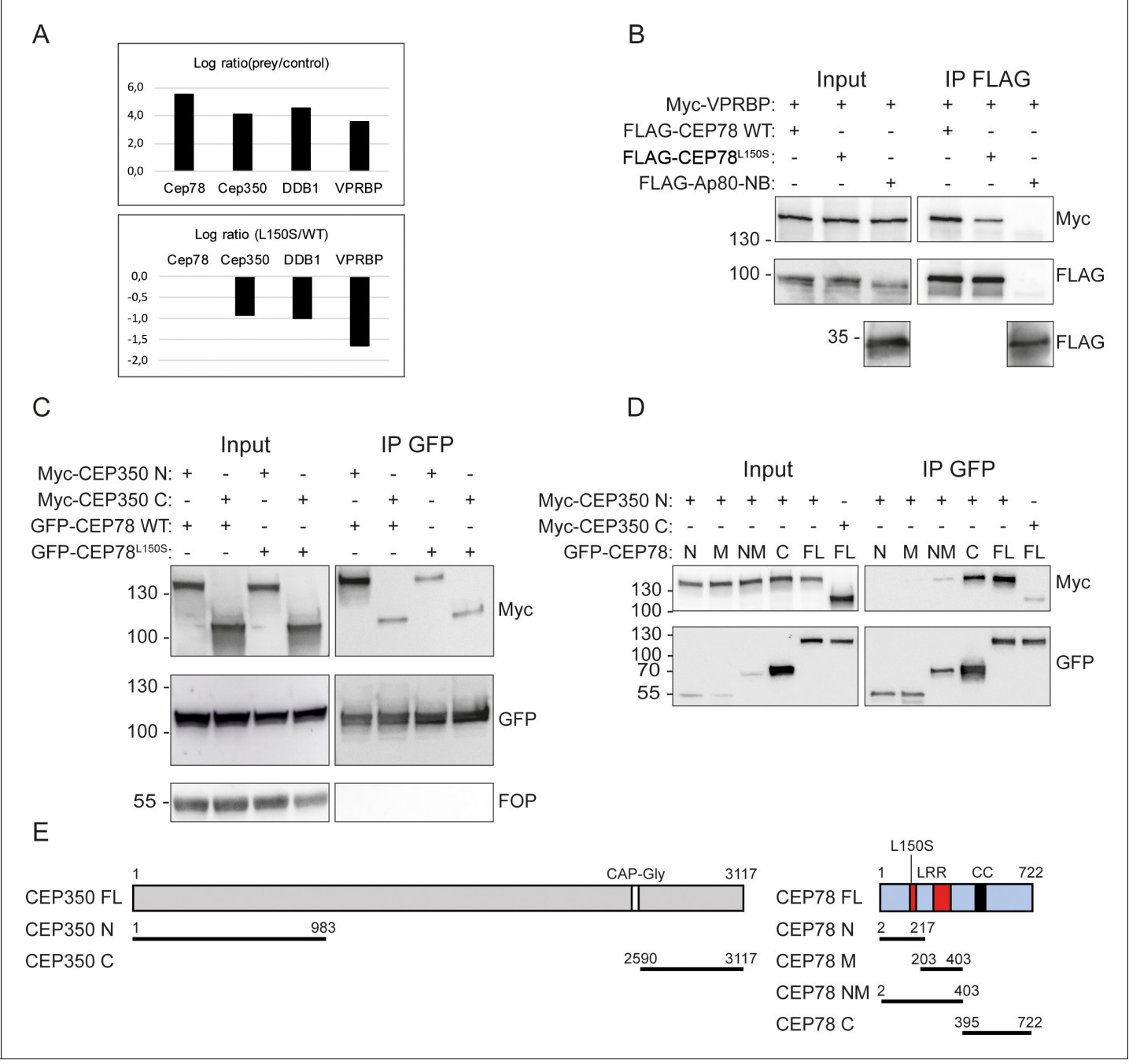

**Figure 3.** CEP78 interacts with the EDD-DYRK2-DDB1[VPRBP] complex and CEP350. (A) Overview of results from CEP78 interactome analysis in 293T cells grown in Stable Isotope Labeling by Amino acids in Cell culture (SILAC) medium. Cells expressing FLAG-CEP78 wildtype (WT), FLAG-CEP78[L150S], or FLAG-Ap80 (negative control) were subjected to FLAG immunoprecipitation (IP) and pellets analyzed by mass spectrometry. The upper panel displays affinity enrichment of Cep78 and interaction partners in FLAG-CEP78 WT cells relative to control cells. The lower panel shows how the CEP78[L150S] mutation weakens these interactions relative to CEP78 WT using ratios normalized for equal affinity enrichment of FLAG-tagged CEP78. (B–D) 293T cells expressing the indicated Myc-, FLAG-, or GFP-fusions were subjected to IP with anti-FLAG (B) or -GFP (C, D) beads and input and pellet fractions analyzed by SDS-PAGE and western blotting with Myc, FLAG, or GFP antibodies as indicated. FLAG-Ap80-NB used in (B) is a negative control. Molecular mass markers are indicated in kDa to the left of the blots. (E) Schematic of the CEP350 and CEP78 fusions used in IP analysis. LRR: leucine-rich repeat; CC: coiled coil. The CEP350 constructs were described in *Eguether et al., 2014*; *Hoppeler-Lebel et al., 2007*; the CEP78 constructs were generated in this study based on published sequence information (*Hossain et al., 2017*; *Ascari et al., 2020*; *Brunk et al., 2016*). N: N-terminal region; M: middle region; NM: N-terminal plus middle region; C: C-terminal region; FL: full length.

The online version of this article includes the following source data and figure supplement(s) for figure 3:

**Source data 1.** Raw data from the CEP78 interactome analysis depicted in *Figure 3A*.

*Figure 3 continued on next page*

## CEP350 regulates the stability and centrosomal recruitment of CEP78

In non-ciliated HeLa and U2OS cells, endogenous CEP78 was reported to localize to both centrioles during all phases of the cell cycle and appeared to concentrate at the centriole wall (*Brunk et al., 2016*). In ciliated RPE1 cells, CEP78 was shown to localize to the distal end of both centrioles, with a preference for the mother centriole/basal body (*Hossain et al., 2017*). Similarly, CEP350 was reported to localize to the distal end of the mother centriole/basal body, near the subdistal appendages, in ciliated RPE1 cells (*Kanie et al., 2017*; *Mojarad et al., 2017*; *Wang et al., 2018*). In agreement with these reports and with our interactome and IP results (*Figure 3*), 3D structured illumination microscopy (SIM) showed partial co-localization of mNG-CEP78 with CEP350 at the distal end of the centrioles in RPE1 cells, adjacent to the distal appendage marker CEP164 and the centriole capping complex protein CP110 (*Figure 4A*). Superimposing fluorescence images on electron micrographs of the mother centriole (*Paintrand et al., 1992*) indicated that mNG-CEP78 localizes to a ring structure of centriole-sized diameter at the distal half of the centriole, encompassed by a wider unclosed ring structure of CEP350. mNG-CEP78 but not CEP350 depicted decreased levels at the daughter centriole as compared to the mother. Furthermore, using a previously characterized RPE1 *CEP350* KO cell line (*Kanie et al., 2017*) that displays significantly reduced centrosomal levels of CEP350 (*Figure 4—figure supplement 1*), we found that depletion of CEP350 not only reduces localization of CEP78 to the centrosome, but also its total cellular level (*Figure 4B–G*). Similarly, we found significantly reduced levels of CEP78 in *FOP* KO cells (*Figure 4B, C*), in line with the decreased centrosomal levels of CEP350 observed in these cells (*Kanie et al., 2017*). These results indicate that CEP350 promotes the recruitment of CEP78 to the mother centriole/basal body as well as its overall stability.

## CEP78 mutant cells display reduced centrosomal levels of EDD1

Next, we asked if CEP78 might affect the centrosomal recruitment of components of the EDD1-DYRK2-DDB1^VPRBP complex and/or CEP350, and analyzed the cellular or centrosomal levels of relevant complex components in RPE1 WT and *CEP78* KO cells. The results showed that serum-deprived, ciliated *CEP78* KO cells display significantly increased centrosomal levels of CEP350 (*Figure 5A, B*) as compared to controls. However, in serum-deprived, non-ciliated cells, the centrosomal levels of CEP350 were not significantly different in *CEP78* KO cells compared to WT controls (*Figure 5C, D*). Similar results were obtained for CEP350 in control and CEP78-deficient patient HSFs (*Figure 5E–H*). We also analyzed the cellular levels of VPRBP by western blotting and the centrosomal levels of VPRBP and EDD1 by IFM in RPE1 WT and *CEP78* KO cells. In agreement with a previous study (*Hossain et al., 2017*), we found that cellular and centrosomal levels of VPRBP were not significantly reduced in serum-fed or serum-deprived *CEP78* KO cells compared to controls (*Figure 6A–C*, *Figure 6—figure supplement 1*). Thus, CEP78 seems to be dispensable for recruitment of VPRBP and CEP350 to the centrosome. In contrast, centrosomal levels of EDD1 were significantly reduced in RPE1 *CEP78* KO cells compared to WT controls, both in ciliated and non-ciliated cells (*Figure 6D–G*). Thus, CEP78 is required for efficient recruitment of EDD1 to the centrosome. To corroborate this result, we measured the centrosomal levels of EDD1 in RPE1 *CEP350* KO cells and found that EDD1 levels were also significantly reduced when compared to the controls (*Figure 6—figure supplement 2*). Together, our data indicate that CEP350, by recruiting CEP78 to the centrosome, also promotes the centrosomal recruitment of EDD1.

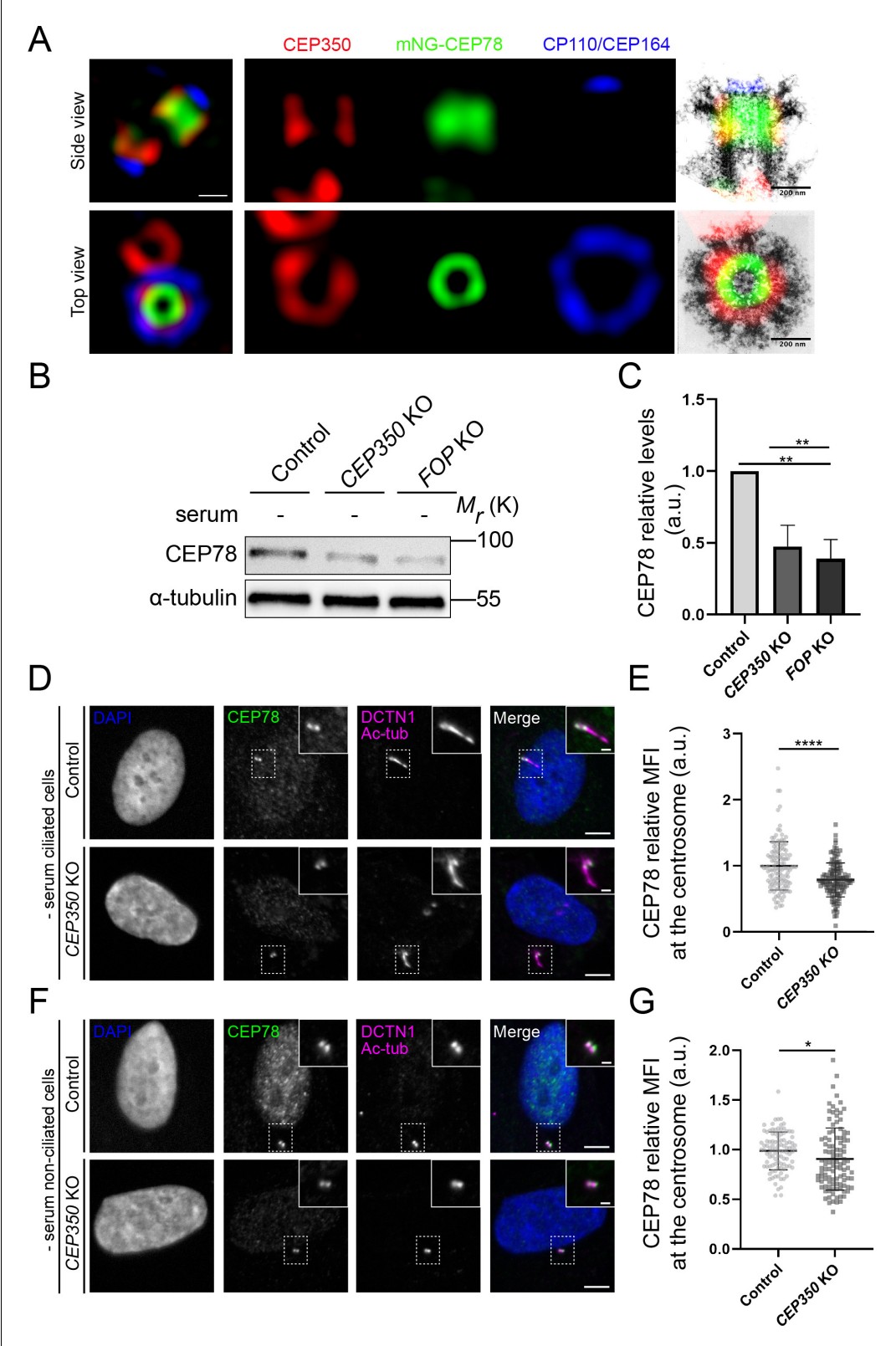

**Figure 4.** CEP350 regulates stability and centrosomal recruitment of CEP78. (**A**) 3D structured illumination microscopy (SIM) analysis of mNG-CEP78 and CEP350 localization in wildtype (WT) RPE1 cells, relative to CP110 and CEP164 (maximal z-projections). Cells were fixed and stained with the indicated antibodies. Montage panels show single-channel images of the mother centriole region (left panels) and overlays on previously published electron micrographs of isolated mother centrioles from KE37 cells, reproduced with permission from *Paintrand et al., 1992* (top: cross-sections of *Figure 4 continued on next page*

Figure 4 continued

proximal and subdistal-appendage region; bottom: side view). Scale bar, 200 nm. (**B**) Western blot analysis of CEP78 and α-tubulin (loading control) in serum-deprived hTERT-RPE1-BFP-Cas9 control (pMCB306), *CEP350* knockout (KO) or *FOP* KO cells (*Kanie et al., 2017*). (**C**) Quantification of data in (**B**), based on three independent experiments analyzed in duplicates. Error bars indicate SD. Statistical analysis was done using Student's t-test (unpaired, two-tailed). (**D, F**) Immunofluorescence microscopy (IFM) analysis of serum-deprived control and *CEP350* KO ciliated (**D**) and non-ciliated cells (**F**) using antibodies as indicated. Dashed boxes indicate cropped images to highlight the centrosomal/ciliary area. Scale bars: 5 μm in representative images and 1 μm in insets. (**E, G**) Quantification of the relative mean fluorescence intensity (MFI) of CEP78 at the centrosome in serum-starved, ciliated (**E**) and non-ciliated (**G**) control and *CEP350* KO cells, based on images as shown in (**D**) and (**F**), respectively. Statistical analysis was done using a two-tailed and unpaired Mann–Whitney test for ciliated cells and two-tailed and unpaired Student's t-test for non-ciliated cells. Accumulated data from three individual experiments (n = 128 and 132 for control and *CEP350* KO ciliated cells, respectively; n = 101 and 107 for control and *CEP350* KO non-ciliated cells, respectively). a.u., arbitrary units; *p<0.05; **p<0.01; ****p<0.0001.

The online version of this article includes the following source data and figure supplement(s) for figure 4:

**Source data 1.** Original western blots for *Figure 4B*.
**Figure supplement 1.** Validation of *CEP350* knockout (KO) cells by immunofluorescence microscopy (IFM) analysis.

## Increased cellular and centrosomal levels of CP110 in CEP78-deficient cells

Since EDD1 is an E3 ubiquitin-protein ligase that negatively regulates CP110 levels (*Hossain et al., 2017*), and whose depletion impairs ciliogenesis (*Kim et al., 2010*; *Shearer et al., 2018*), we asked if the loss of CEP78 would lead to altered levels of CP110 at the centrosome. Indeed, western blot and IFM analysis of RPE1 WT and *CEP78* KO cells revealed that total cellular and centrosomal CP110 levels are significantly increased in *CEP78* KO cells compared to WT controls, both under serum-deprived (*Figure 7A–D, F, G*, *Figure 7—figure supplement 1A, B*) and serum-fed conditions (*Figure 7—figure supplement 2A–D*), whereas CP110 levels appeared normal upon stable expression of mNG-CEP78 in the *CEP78* KO background (*Figure 7—figure supplement 1C*). We also analyzed CP110 levels specifically at the mother centriole of ciliated RPE1 WT and *CEP78* KO cells and found no significant differences between them (*Figure 7E*). Similar results were obtained using *CEP78* mutant and control HSF cells (*Figure 7G, H*). Since a previous study reported that siRNA-mediated depletion of CEP78 in RPE1 and HeLa cells reduces the cellular levels of CP110 (*Hossain et al., 2017*), a result contradicting our own observations (*Figure 7*, *Figure 7—figure supplements 1* and *2*), we tested if the elevated CP110 levels seen in our *CEP78* mutant cells might be due to compensatory upregulation of CP110 mRNA expression. However, RNA-seq analysis indicated no significant changes in CP110 mRNA levels in patient-derived, *CEP78*-deficient HSF cultures compared to controls; mRNA levels of CEP350, VPRBP, and DDB1 in *CEP78*-deficient HSF cultures were also similar to those of controls (*Figure 7—figure supplement 3*). Furthermore, a cycloheximide chase experiment indicated that CP110 is more stable upon serum deprivation of *CEP78* KO cells compared to WT controls (*Figure 7—figure supplement 4*). Taken together, we conclude that loss of CEP78 leads to elevated cellular and centrosomal levels of CP110 and that this is likely due to increased stability of CP110. Furthermore, our results indicate that in ciliated *CEP78* KO cells, the amount of CP110 present specifically at the mother centriole is not significantly higher than in control cells.

## Partial depletion of CP110 normalizes ciliation frequency of *CEP78* KO cells

CP110 is a key negative regulator of ciliogenesis (*Spektor et al., 2007*; *Tsang and Dynlacht, 2013*) that was also shown to be required for ciliogenesis initiation as well as ciliary length control (*Yadav et al., 2016*; *Walentek et al., 2016*). The above results therefore prompted us to test if the reduced ciliation frequency and/or increased length of remaining cilia observed in *CEP78*-deficient cells could result from elevated centrosomal CP110 levels in these cells. When we used siRNA to partially deplete cellular and centrosomal CP110 from RPE1 WT and *CEP78* KO cells and subjected the cells to serum deprivation (*Figure 8A, B*, *Figure 8—figure supplement 1*), we found that the *CEP78* KO cells restored their ciliation frequency to WT levels, whereas partial CP110 depletion had little effect on cilia numbers in WT cells grown under similar conditions (*Figure 8C, D*). In contrast, depletion of CP110 from *CEP78* KO cells did not rescue the ciliary length defect of residual cilia; these cilia were still significantly longer than those of WT controls (*Figure 8C, E*). Thus, partial

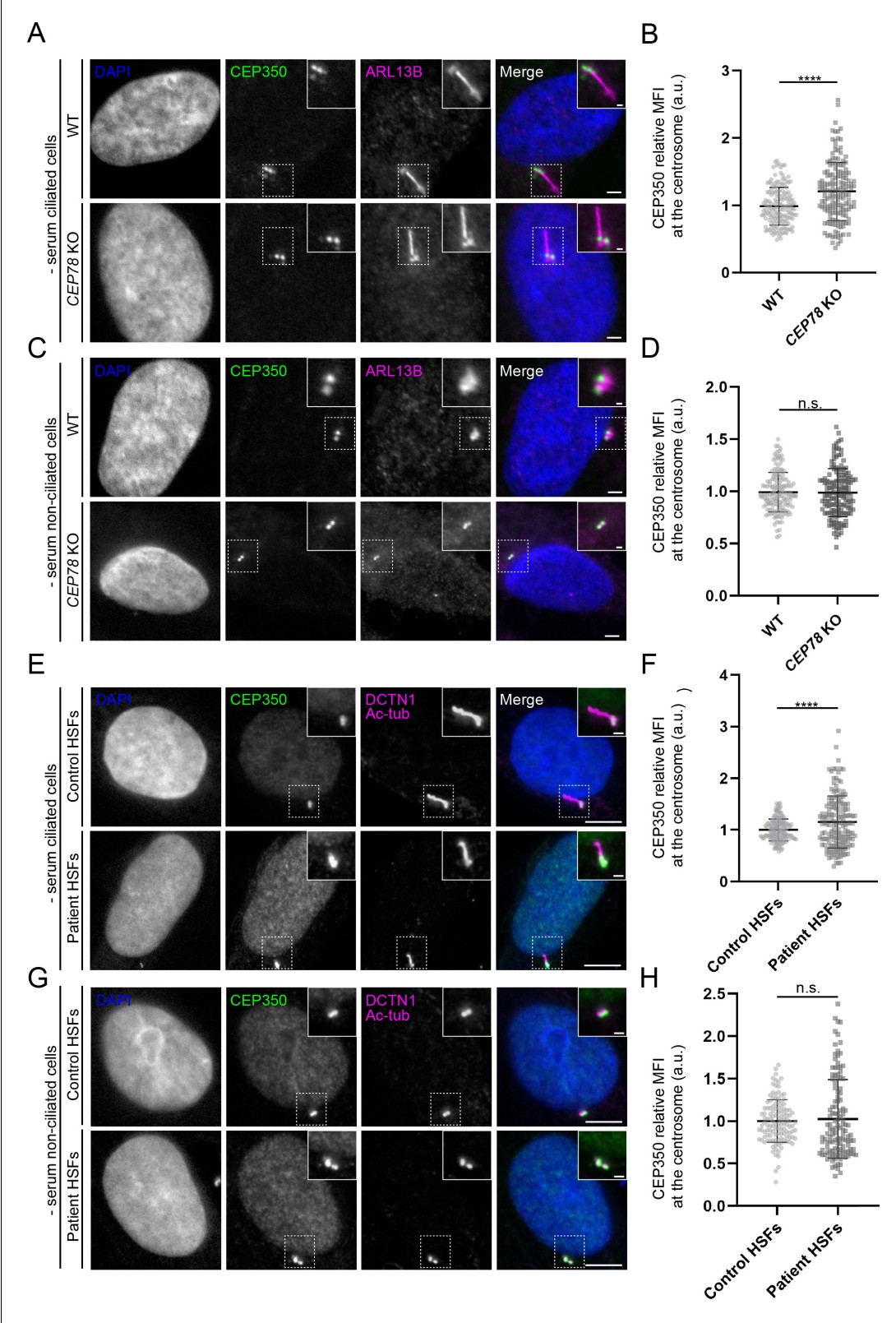

**Figure 5.** Altered centrosomal levels of CEP350 in *CEP78* mutant cells. (**A, C, E, G**) Representative immunofluorescence microscopy (IFM) images of ciliated (**A**) and non-ciliated (**C**) serum-deprived RPE1 wildtype (WT) and *CEP78* knockout (KO) cells, and ciliated (**E**) and non-ciliated (**G**) serum-deprived control and CEP78-deficient (Patient) human skin fibroblasts (HSFs) (combined data from HSFs from patient F3: II:1 [*Ascari et al., 2020*] and patient 2702 r34, individual II-3 [*Nikopoulos et al., 2016*]) labeled with antibodies indicated in the figure. Insets show closeups of the centrosomal/ciliary area
*Figure 5 continued on next page*

*Figure 5 continued*

(dashed boxes). Scale bars: 5 µm in representative images and 1 µm in closeups. (**B, D**). Quantification of CEP350 relative mean fluorescence intensity (MFI) at the centrosome based on images as shown in (**A, C**). Data is shown as mean ± SD. A Student's t-test was used as statistical analysis between the two groups based on three individual experiments (n = 157 or 158 for ciliated cells and n = 144–152 for non-ciliated cells). (**F, H**) Quantification of CEP350 relative MFI at the centrosome in HSFs based on images as shown in (**E, G**). Data is shown as mean ± SD. Student's t-test was used as statistical analysis between the two cell groups based on seven individual experiments with a total of 154 and 156 analyzed ciliated cells (**F**), and three individual experiments with a total of 122 or 126 analyzed non-ciliated cells (**H**). a. u., arbitrary units; n.s., not statistically significant; ****p<0.0001.

depletion of CP110 from *CEP78* KO cells can rescue their reduced ciliation frequency phenotype but not the increased length of remaining cilia. This result is in line with our observation that a *CEP78* KO clone expressing a truncated version of CEP78 (clone #44) displays significantly reduced ciliation frequency, but normal length of the cilia that do form (*Figure 1—figure supplement 1*), implying that CEP78 regulates ciliogenesis and ciliary length by separate mechanisms.

## Discussion

Several studies have shown that loss-of-function mutations in *CEP78* are causative of ciliopathy characterized by CRDHL (*Nikopoulos et al., 2016*; *Ascari et al., 2020*; *Namburi et al., 2016*; *Fu et al., 2017*), and that cells lacking CEP78 display reduced ciliation frequency (*Azimzadeh et al., 2012*) or abnormally long cilia (*Nikopoulos et al., 2016*; *Ascari et al., 2020*). However, the molecular mechanisms by which CEP78 regulates cilium biogenesis and length were so far unclear. In this study, we found that loss of CEP78 leads to decreased ciliation frequency as well as increased length of the residual cilia that do form, both in RPE1 cells and patient-derived HSFs. Rescue experiments in *CEP78* KO RPE1 cells stably expressing mNG-CEP78 or mNG-CEP78$^{L150}$ supported that the observed phenotypes were specifically caused by loss of CEP78 at the centrosomes. Interestingly, we found that both ciliated and non-ciliated CEP78-deficient cells fail to efficiently recruit EDD1 to the centrosome, whereas cellular and overall centrosomal levels of CP110 are dramatically increased. However, in the subpopulation of *CEP78* KO cells that are ciliated, CP110 levels specifically at the mother centriole were similar to those of ciliated control cells, implying that the elevated centrosomal CP110 level observed in these cells is caused by a pool of CP110 located at the daughter centriole and/or pericentriolar area. This result also indicates that the abnormally long cilia phenotype seen in the *CEP78* KO cells is not caused by increased CP110 levels at the mother centriole. Supportively, siRNA-mediated depletion of CP110 could restore cilia frequency, but not length, to normal in the *CEP78* KO cells. Furthermore, a *CEP78* mutant line, clone #44, expressing a shorter version of CEP78 likely lacking the N-terminus displayed reduced ciliation frequency but normal length of residual cilia, substantiating that CEP78 regulates ciliogenesis and cilia length by distinct mechanisms. Specifically, our results indicate that CEP78 promotes ciliogenesis by negatively regulating CP110 levels at the mother centriole via activation of VprBP, but negatively regulates ciliary lengthening independently of CP110 (*Figure 8F*).

The removal of CP110 from the distal end of the mother centriole constitutes a key step in the initiation of ciliogenesis and is regulated by a growing number of proteins that include TTBK2 (*Goetz et al., 2012*) and components of the UPS system such as Neurl-4 (*Loukil et al., 2017*) and the cullin-3 (CUL3)–RBX1–KCTD10 complex (*Nagai et al., 2018*). The EDD1-DYRK2-DDB1$^{VPRBP}$ complex was similarly shown to mediate degradation of CP110 and to interact with CEP78 (*Hossain et al., 2017*), but the precise consequences of these activities for ciliogenesis were unclear. Our work strongly suggests that CEP78 functions together with the EDD1-DYRK2-DDB1$^{VPRBP}$ complex to negatively regulate centrosomal CP110 levels at the onset of ciliogenesis, leading to its removal from the distal end of the mother centriole (*Figure 8F*). This is in contrast to a previously proposed model, which suggested that CEP78 inhibits degradation of CP110 by the EDD1-DYRK2-DDB1$^{VPRBP}$ complex (*Hossain et al., 2017*). The reason for this discrepancy is unclear, but may be due to different experimental approaches used in our study and the paper by *Hossain et al., 2017*. In this latter study, siRNA was used to partially deplete CEP78 from cultured cells, whereas we performed our experiments on cells with endogenous *CEP78* mutations. CP110 removal from the distal end of the mother centriole occurs concomitantly with docking of the centriole to vesicles or the

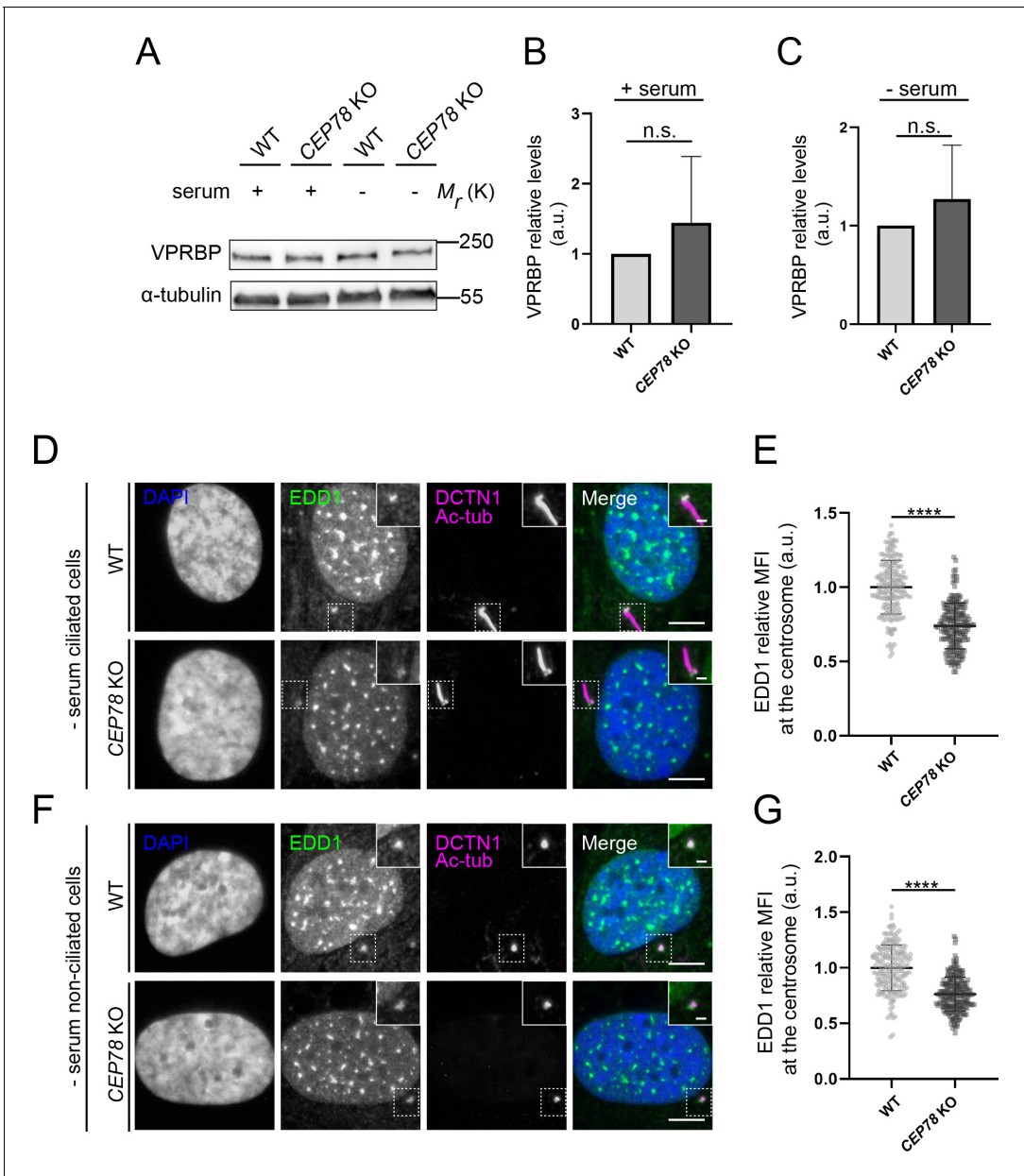

**Figure 6.** Analysis of EDD-DYRK2-DDB1$^{VPRBP}$ complex components in CEP78-deficient cells. (A) Western blot analysis of VPRBP in lysates from serum-fed and serum-deprived RPE1 wildtype (WT) and *CEP78* knockout (KO) cells. α-tubulin was used as a loading control. (B, C) Quantification of the VPRBP relative levels in the different conditions depicted in (A). Statistical analysis was performed using a Student's t-test (unpaired, two-tailed) from five independent experiments analyzed in duplicates. Error bars indicate SD. (D, F) Representative immunofluorescence microscopy (IFM) images of ciliated (D) and non-ciliated (F) serum-deprived RPE1 WT and *CEP78* KO cells labeled with antibodies against EDD1 (green) and DCTN1 plus acetylated tubulin (magenta). DAPI was used to mark the nucleus (blue). Insets show enlarged views of the cilium-centrosome region. Scale bars: 5 μm in original images and 1 μm in closeups. (E, G) Quantification of the EDD1 mean fluorescence intensity (MFI) at the centrosome based on images as shown in (D) and (F) using a two-tailed and unpaired Student's t-test. Data is shown as mean ± SD. Student's t-test from three independent experiments (n = 194 and n = 201 for ciliated WT and *CEP78* KO cells, respectively; n = 194 and n = 217 for non-ciliated WT and *CEP78* KO cells, respectively). Data is shown as mean ± SD. a.u., arbitrary units; n.s., not statistically significant; ****p<0.0001.

The online version of this article includes the following source data and figure supplement(s) for figure 6:

**Source data 1.** Original western blots for *Figure 6A*.

**Figure supplement 1.** VPRBP centrosomal levels are increased or unchanged in RPE1 *CEP78* knockout (KO) cells.

**Figure supplement 2.** EDD1 centrosomal levels are reduced in RPE1 *CEP350* knockout (KO) cells.

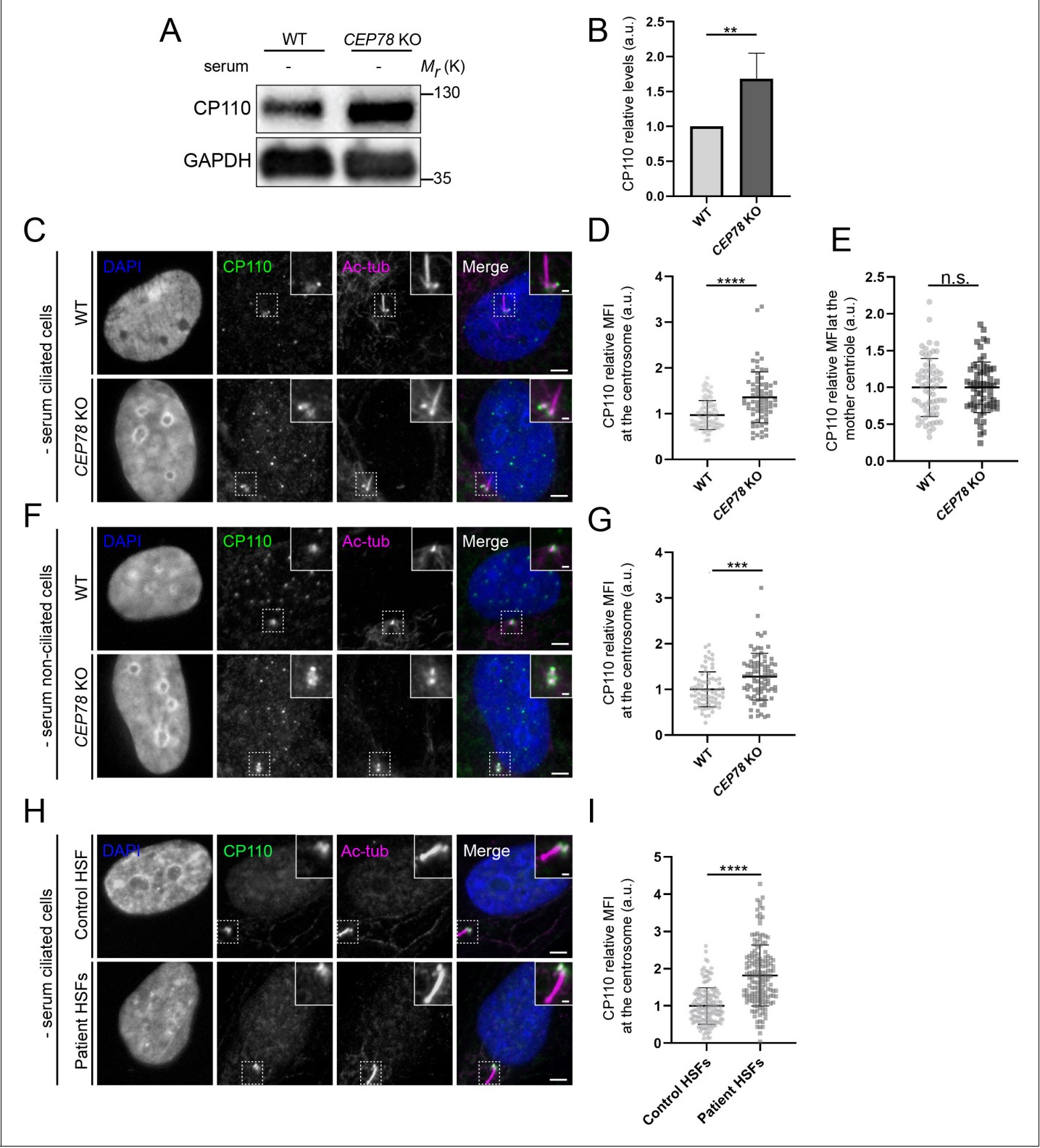

**Figure 7.** CEP78-deficient cells display elevated cellular and centrosomal levels of CP110. (**A**) Western blot analysis of lysates from serum-deprived RPE1 wildtype (WT) and *CEP78* knockout (KO) cells using indicated antibodies. GAPDH was used as a loading control. (**B**) Quantification of the data shown in (**A**), based on four independent experiments analyzed in duplicates. Error bars indicate SD. Statistical analysis was done using Student's t-test (unpaired, two-tailed). (**C, F**) Representative immunofluorescence microscopy (IFM) images of serum-starved ciliated (**C**) and non-ciliated (**F**) RPE1 WT and *CEP78* KO cells labeled with antibodies against CP110 (green), combined DCTN1 and acetylated tubulin (Ac-tub; magenta) and DAPI to mark the

*Figure 7 continued on next page*

Figure 7 continued

nucleus (blue). Insets show enlarged views of the cilium-centrosome area. Scale bars: 5 µm in original images and 1 µm in closeups. (D, G) Quantification of the relative mean fluorescence intensity (MFI) of CP110 at the centrosome based on images as shown in (C) and (F), respectively. Student's t-test (unpaired, two-tailed) from three independent biological experiments (n = 90 and n = 75 for ciliated RPE1 WT and *CEP78* KO cells, respectively; n = 75 and n = 82 for non-ciliated RPE1 WT and CEP78 KO cells, respectively) was used for statistical analysis. Data is presented as mean ± SD. (E) Quantification of the relative MFI of CP110 at the mother centriole based on images shown in (C). Mann–Whitney test (two-tailed and unpaired) was used as statistical analysis based on two independent experiments (n = 61 and n = 62 for RPE1 WT and CEP78 KO cells, respectively). Data is presented as mean ± SD. (H) Representative IFM images of serum-deprived healthy and CEP78-deficient (Patient) human skin fibroblasts (HSFs) (data from patient 2702 r34, individual II-3 described in *Nikopoulos et al., 2016*) labeled with the indicated antibodies. DAPI was used as counterstaining to mark the nucleus. Dashed lines show closeup images of the centrosome region. Scale bars: 5 µm in original images and 1 µm in closeups. (I) Quantification based on observations of 151 and 155 cells of CEP78 control and patient cells, respectively, from three individual experiments. Student's t-test (unpaired and two-tailed) was used to assess the differences between the two groups. a.u., arbitrary units; n.s., not statistically significant; **p<0.01; ***p<0.001; ****p<0.0001.

The online version of this article includes the following source data and figure supplement(s) for figure 7:

**Source data 1.** Original western blots for *Figure 7A*.
**Figure supplement 1.** Elevated cellular CP110 levels in RPE1 *CEP78* knockout (KO) clones.
**Figure supplement 1—source data 1.** Original western blots for *Figure 7—figure supplement 1*.
**Figure supplement 2.** Serum-fed CEP78-deficient cells display elevated cellular and centrosomal levels of CP110.
**Figure supplement 2—source data 1.** Original western blots for *Figure 7—figure supplement 2A*.
**Figure supplement 3.** Barplots showing expression of *CP110, CEP350, VRPBP,* and *EDD1* genes for patients (n = 3) and controls (n = 4), based on RNA-seq data of skin fibroblasts.
**Figure supplement 3—source data 1.** Raw RNA-seq data for *Figure 7—figure supplement 3*.
**Figure supplement 4.** Relative CP110 levels in wildtype (WT) and *CEP78* knockout (KO) cells treated with cycloheximide.
**Figure supplement 4—source data 1.** Original western blots for *Figure 7—figure supplement 4A*.

plasma membrane at the onset of ciliogenesis (*Shakya and Westlake, 2021*). Therefore, a role for CEP78 in removal of CP110 from the centrosome is compatible with ultrastructural analyses in *Planarians*, which indicated that basal bodies fail to dock to the plasma membrane in CEP78 RNAi-depleted animals (*Azimzadeh et al., 2012*). Although not addressed in detail in the current study, the elevated centrosomal CP110 levels observed in serum-fed CEP78-deficient cells are also compatible with previous work implicating CEP78 in regulation of PLK4-mediated centriole duplication (*Brunk et al., 2016*).

We also uncovered a new physical interaction between CEP78 and the N-terminal region of CEP350 and found that CEP350 is important for recruitment of CEP78 to the centrosome as well as for its overall stability. The reduced interaction of the CEP78$^{L150S}$ mutant protein with CEP350 might explain its lack of detection by western blot analysis of patient HSFs (*Ascari et al., 2020*) since reduced interaction with CEP350 is likely to decrease the long-term stability of CEP78$^{L150S}$. Our IP and IFM results furthermore suggest that CEP350 controls CP110 removal from the mother centriole not only via recruitment of TTBK2 (*Kanie et al., 2017*), but also via CEP78-dependent recruitment of EDD1. Indeed, we observed reduced centrosomal levels of EDD1 in *CEP350* KO cells, consistent with this idea. It remains to be determined whether these two pathways act in sequence or in parallel to control centrosomal CP110 levels. Furthermore, as the recruitment of CEP350 to centrioles itself depends on FOP (*Kanie et al., 2017*; *Mojarad et al., 2017*; *Yan et al., 2006*) and on the distal centriole protein C2CD3 (*Huang et al., 2018*), loss of these proteins may similarly affect the centrosomal recruitment of CEP78 and EDD1, and thereby negatively regulate CP110 levels via this pathway. It will be important to investigate these possibilities in future studies.

While increased CP110 levels can account for the decreased ciliation frequency observed in serum-deprived CEP78 mutant cells, it remains unclear why some of these cells form abnormally long cilia. Our analysis indicated that the ciliated mutant cells are able to locally displace or degrade CP110 at the mother centriole, but the underlying mechanism remains to be determined. Of note, the *CEP350* KO cells used in our study also seem to display a similar dual phenotype since approximately 30% of these cells form cilia despite an inability to recruit TTBK2 and remove CP110 from the mother centriole (*Kanie et al., 2017*). Moreover, a recent genome-wide siRNA knockdown analysis showed that depletion of DDB1 caused an elongated cilia phenotype, whereas depletion of EDD1 (UBR5) leads to fewer but normal length cilia (*Wheway et al., 2015*), even though DDB1 and EDD1 are part of the same E3 ubiquitin ligase complex (*Nakagawa et al., 2013*). Thus, a slight imbalance

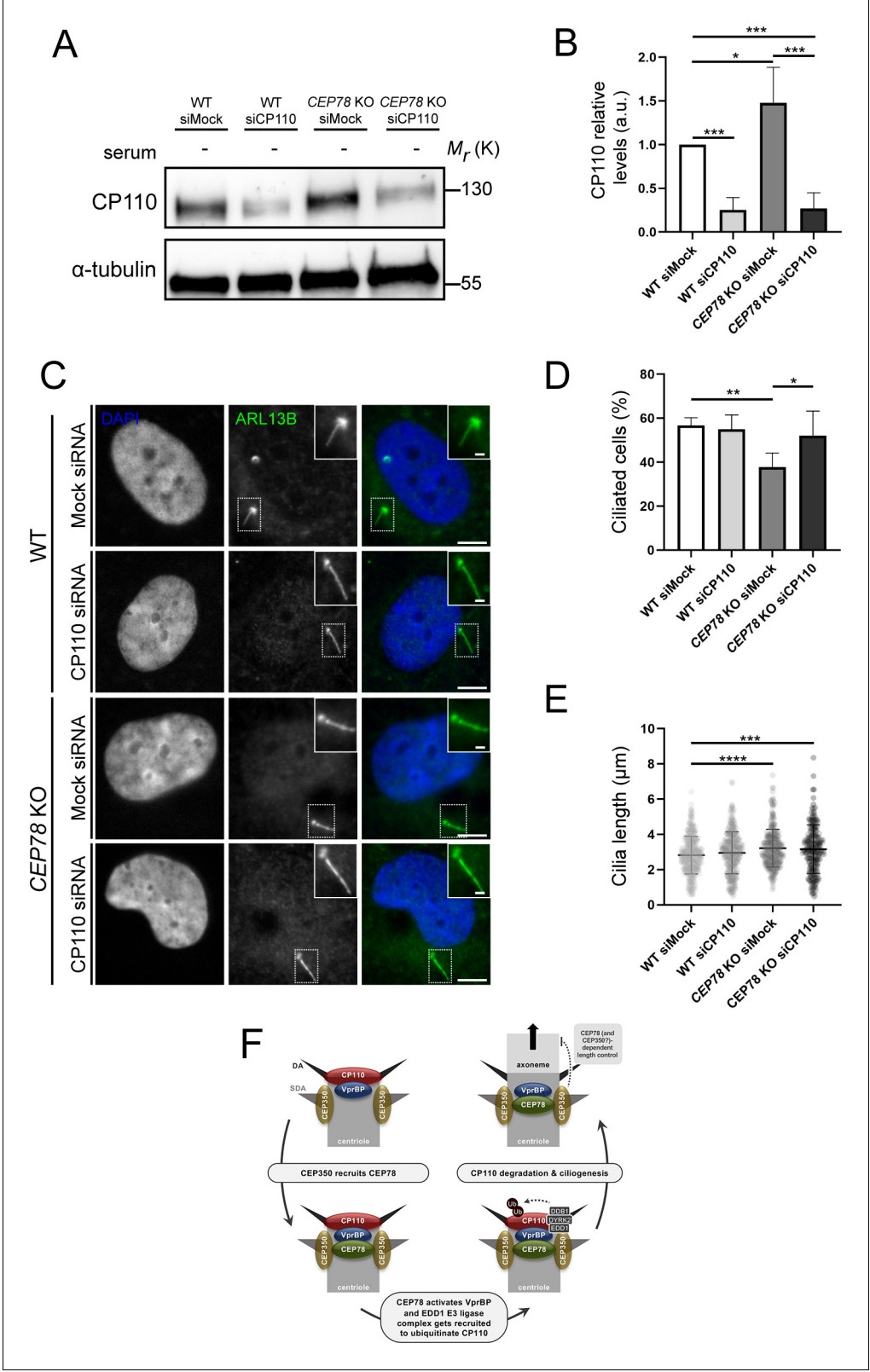

**Figure 8.** Depletion of CP110 from *CEP78* knockout (KO) cells rescues their ciliation frequency phenotype. (**A**) Representative western blots of RPE1 wildtype (WT) or *CEP78* KO cells treated with control (siMock) or CP110-specific siRNA (siCP110). Cells were deprived of serum and blots were probed with antibody against CP110 or α-tubulin (loading control). (**B**) Quantification of relative cellular CP110 levels under different conditions based on

*Figure 8 continued*

blots as shown in (A) (n = 5 for WT and n = 4 for *CEP78* KO cells; each sample was analyzed in duplicates). Error bars indicate SD. Data was normalized in relation to the WT cells transfected with Mock siRNA (WT siMock), designated as the control group. Differences of CP110 relative levels between *CEP78* KO cells transfected with Mock or CP110 siRNA and WT cells transfected with CP110 siRNA in relation to control were assessed by performing an ordinary one-way ANOVA with a Dunnet's multiple comparison test. Differences of CP110 cellular levels between *CEP78* KO cells transfected with the same siRNAs were addressed using an unpaired and two-tailed Student's t-test. (C) Representative immunofluorescence microscopy (IFM) images of serum-deprived RPE1 WT and *CEP78* KO cells treated with control (siMock) or CP110-specific siRNA (siCP110) and stained for ARL13B (green). DAPI marks the nucleus. Dashed boxes indicate cropped images to highlight the centrosomal/ciliary zone. Scale bars, 5 µm in representative images and 1 µm in insets. (D, E) Quantification of percentage of ciliated cells (D) and length of remaining cilia (E) in RPE1 WT or *CEP78* KO cells treated with control (siMock) or CP110-specific siRNA (siCP110). The quantification is based on images as shown in (C) and represents data from four and five independent experiments, respectively, for RPE1 WT (n = 554–617 cells per condition) and *CEP78* KO cells (n = 846–856 cells per condition). Ordinary one-way ANOVA with Dunnett's multiple comparison test was used to compare the mean of the remaining groups in relation with the mean of RPE1 WT cells treated with siMock, which was designated as the control group. Differences between *CEP78* KO cells transfected with the above-mentioned siRNA were discriminated using an unpaired and two-tailed Student's t-test. Bars in (D) represent SD and data in (E) is presented as mean ± SD. a.u., arbitrary units; *p<0.05; **p<0.01; ***p<0.001; ****p<0.0001. (F) Proposed model for how CEP78 regulates ciliogenesis and ciliary length control. See text for details.

The online version of this article includes the following source data and figure supplement(s) for figure 8:

**Source data 1.** Original bots for *Figure 8A*.

**Figure supplement 1.** Validation of CP110 knockdown in RPE1 wildtype (WT) and *CEP78* knockout (KO) cells by immunofluorescence microscopy (IFM).

in the regulation of this complex might determine whether or not cilia are formed and how long they are. It should also be noted that some studies have reported dual roles for CP110 during ciliogenesis, that is, CP110 may not only function to inhibit ciliogenesis but may also promote ciliary extension in some contexts (*Yadav et al., 2016*; *Walentek et al., 2016*). Although the mechanism responsible for the elongated cilia phenotype seen in some of the CEP78 mutant cells remains unclear, one attractive candidate to be involved is CEP350, which we found to be present at abnormally high levels at the centrosome of ciliated *CEP78* KO cells, specifically. Consistent with this idea, we have observed that the CEP78 truncation mutant (clone #44), which expresses fewer, but normal length cilia, displays normal or slightly reduced levels of CEP350 at the centrosome in ciliated as well as in non-ciliated cells (data not shown). Alternatively, one or more substrates of EDD1 other than CP110 could also be affected in the *CEP78* mutant cells, thereby affecting ciliary length control. For example, EDD1 was also shown to mediate ubiquitination of CSPP-L in turn promoting its recruitment to centriolar satellites and the centrosome (*Shearer et al., 2018*). CSPP-L is a well-known positive regulator of primary cilia formation (*Patzke et al., 2010*) that was recently shown to bind directly to CEP104 to control axoneme extension (*Frikstad et al., 2019*). Furthermore, CEP104 is implicated in ciliogenesis through interaction with the CEP97-CP110 complex and modulation of microtubule dynamics at the ciliary tip (*Frikstad et al., 2019*; *Jiang et al., 2012*; *Satish Tammana et al., 2013*). It will therefore be interesting to investigate if and how CSPP-L and CEP104 localization is affected in CEP78 mutant cells. Moreover, as CEP350 additionally binds to CYLD (*Eguether et al., 2014*), a deubiquitinase that controls centriolar satellite homeostasis and ciliogenesis through de-ubiquitination of the E3 ubiquitin ligase MIB1 (*Wang et al., 2016*; *Douanne et al., 2019*), and was proposed to regulate axonemal IFT particle injection through FOP-CEP19-RABL2 interactions (*Kanie et al., 2017*; *Nishijima et al., 2017*; *Mojarad et al., 2017*), it is tempting to speculate that CEP78 might impinge on these processes.

## Materials and methods

### PCR and cloning

For the establishment of the N-terminal FLAG-tagged CEP78 construct, a cDNA clone coding for human full-length CEP78 cDNA (GeneCopoiea pShuttleGateway PLUS ORF, NCBI accession number

NM_001098802) was used as template for amplification with the Phusion High-Fidelity DNA polymerase (New England BioLabs) under standard PCR conditions and attB forward and reverse primers (see Key resources table) compatible with the Gateway cloning technology (Invitrogen/Thermo Fisher Scientific) for inclusion of the insert into the pDonor201 Gateway vector (Invitrogen/Thermo Fisher Scientific). To produce human FLAG-CEP78$^{L150S}$, site-directed mutagenesis was performed with QuickChange Lightning Site-Directed Mutagenesis Kit (Agilent) and primers CEP78 mut_F and CEP78 mut_R (see Key resources table). Accuracy of the *CEP78* ORF was assessed by Sanger sequencing. Subsequently, the desired fragment was cloned into p3xFLAG-CMV/DEST (gift from Dr. Ronald Roepman lab, Radboud University Medical School, Nijmegen, NL; see Key resources table for details) to produce p3xFLAG-CEP78 or p3xFLAG-CEP78$^{L150S}$.

Plasmids encoding full-length and truncated versions (N-terminus, middle, N-terminus plus middle, C-terminus) of CEP78 tagged with GFP in the N-terminus were generated by PCR using relevant primers (see Key resources table) and pFLAG-CEP78 as template; PCR products were cloned into pEGFP-C1 by standard approaches, following digestion with BamHI (Roche, cat# 1056704001) and KpnI (Roche, cat#10899186001) and ligation with T4 DNA ligase (Applichem, cat# A5188). Ligated plasmids were transformed into competent *Escherichia coli* DH10B cells and cells harboring recombinant plasmids selected on Luria Bertani (LB; Sigma-Aldrich) agar plates with 50 µg/ml kanamycin. Plasmids were purified using Plasmid DNA Mini Kit I (Omega Biotech, cat# D6943-02) or Nucleobond Xtra Midi kit (Macherey-Nagel, cat# 740410.50), according to protocols supplied by the manufacturers. Sequences of plasmid inserts were verified by Eurofins Genomics.

Gateway system-compatible pENTR vectors encoding for mNeonGreen-CEP78 or mNeon-Green-CEP78L150S fusions were generated by subcloning of ORFs from pEGFP constructs described above in frame with mNeonGreen using the BamH1/Kpn1 sites. For lentivirus particle production, these plasmids were recombined with pCDH-EF1a-Gateway-IRES-BLAST plasmids as described in *Frikstad et al., 2019* using LR Clonase II (Invitrogen/Thermo Fisher Scientific).

## Mammalian cell culture and transfection

HSFs were cultured as described previously (*Nikopoulos et al., 2016*; *Ascari et al., 2020*). Human embryonic kidney-derived 293T cells were cultured and transfected with plasmids as described previously (*Schou et al., 2015*). RPE1 cells were grown in T-75 flasks in a 95% humidified incubator at 37°C with 5% CO$_2$. Cells were cultured in Dulbecco's Modified Eagle Medium (DMEM) (GIBCO, cat# 41966-029) supplemented with 10% fetal bovine serum (FBS) and 1% penicillin/streptomycin (P/S). When cells were about 80–90% confluent, they were passaged and setup for new experiments. Upon seeding, RPE1 cells were washed once with preheated 1× phosphate buffered saline (PBS) and then incubated with 1% Trypsin-EDTA (ethylenediaminetetraacetic acid) (Sigma-Aldrich, cat# T4174) solution for 5 min at 37°C. The detached cells were aspired in an appropriate volume of new preheated DMEM and seeded into Petri dishes with or without sterile glass coverslips (*Verdier et al., 2016*). When the cells had reached 80–90% confluency, they were serum starved for 24 hr to induce formation of primary cilia. A small amount of cell solution was passaged on in a new T-75 flask containing new preheated DMEM. All cell lines were tested negative for *Mycoplasma*, and those not generated in this study have been used in prior publications. New cell lines generated in this study were validated by Sanger sequencing.

For transfection with plasmids, RPE1 cells were seeded in 60 mm Petri dishes as described above to about 50–70% confluency at the time of transfection. For rescue experiments, 2 µg pFLAG-CEP78 or pFLAG-CEP78$^{L150S}$ and 6 µl FuGENE6 were mixed in 100 µl serum- and antibiotic-free DMEM and incubated for 20 min at room temperature. New DMEM was added to the cells prior to transfer of the transfection reagent, which was gently dripped onto the cells and swirled to make sure it was dispersed well. After cells were incubated with transfection medium for 4 hr, the medium was replaced with serum-free DMEM for 24 hr to induce growth arrest.

Gene silencing in RPE1 cells by siRNA was performed using DharmaFect (see Key resources table). Briefly, RPE1 cells were seeded in DMEM with serum and antibiotics in 30 mm Petri dishes at 20–25% confluency. Transfection was performed by combining 50 nM siRNA with 5 µl of DharmaFect in 200 µl of serum- and antibiotic-free DMEM for 15 min at room temperature. After incubation, the complexes were added dropwise onto the cells and the media was swirled gently to ensure good dispersion prior to incubation at 37°C. Cells were kept under these conditions for 48 hr and then serum starved for 24 hr before being processed for further experiments.

RPE1 cell transduction with lentivirus particles and Blasticidine selection (10 µg/ml f.c.) was conducted as described in *Frikstad et al., 2019*.

## Immunofluorescence microscopy and image analysis

Standard epifluorescence IFM analysis was performed as described previously (*Verdier et al., 2016*). Briefly, glass coverslips containing cells of interest were washed in PBS and fixed with either 4% PFA solution for 15 min at room temperature or on ice, or with ice-cold methanol for 10–12 min at −20°C. After three washes in PBS, the PFA-fixed cells were permeabilized 10 min in 0.2% Triton X-100% and 1% bovine serum albumin (BSA) in PBS and blocked in 2% BSA in PBS for 1 hr at room temperature. Coverslips were incubated overnight at 4°C in primary antibodies (see Key resources table) diluted in 2% BSA in PBS. The next day, coverslips were washed three times with PBS for 5 min and incubated for 1 hr at room temperature in secondary antibodies (see Key resources table) diluted in 2% BSA in PBS. Coverslips were washed three times in PBS for 5 min before staining with 2 µg/ml DAPI solution PBS for 30 s. Finally, coverslips were washed once with PBS and before mounting on objective glass with 6% n-propyl gallate diluted in glycerol and 10× PBS and combined with Shandon Immuno-Mount (Thermo Scientific, cat# 9990402) in a 1:12 ratio. Cells were imaged on a motorized Olympus BX63 upright microscope equipped with a DP72 color, 12.8 megapixel, 4140 × 3096 resolution camera. cellSense Dimension software 1.18 from Olympus was used to measure cilia length; ImageJ version 2.0.0-rc-69/1.52i was used to measure the relative mean fluorescence intensity (MFI) of relevant antibody-labeled antigens at the centrosome/basal body. A fixed circle was drawn around a centrosome in a cell. This same region of interest (ROI) was used to measure the MFI of a specific protein at the centrosome. A constant ROI was also drawn to measure the cell background signal, which was subtracted from the MFI measured at the centrosome/basal body. Images were prepared for publication using Adobe Photoshop and Adobe Illustrator.

Super-resolution imaging of RPE1 cells was conducted as described in *Frikstad et al., 2019* except that cells were mounted in ProLong Diamond Antifade mountant. Settings for hardware-assisted axial and software-assisted lateral channel alignment and image reconstruction were validated by imaging of RPE1 cells stained for CEP164 and labeled with secondary antibodies conjugated to either Alexa488, DyeLight550, or Alexa647. For superimposing 3D-SIM images on electron microscopy micrographs of centrioles (kindly provided by Dr. Michel Bornens; *Paintrand et al., 1992*), 3D SIM images were sized to same digital pixel resolution as original EM images using the bicubic algorithm in ImageJ and maximal intensity z-projections of single-channel images overlaid on micrographs using centriole centers/centriole distal ends as reference points.

## Statistical analysis

Statistical analyses were performed using GraphPad Prism 6.0. The background-corrected MFI measured at the centrosome was normalized to relevant control cells. Mean and standard deviation (SD) was calculated for all groups, and outliers were identified and removed by the ROUT method before the statistical tests were conducted. The data was tested for Gaussian normality using either D'Agostino's K-squared test or Shapiro–Wilk test. Depending on the distribution of the data, two-tailed and unpaired Student's t-test or Mann–Whitney test were used when comparing two groups. Also, depending on the distribution of the data, one-way ANOVA followed by Tukey's, Dunnet's or Dunn's multiple comparison tests was used when comparing more than two groups. Unless otherwise stated, the statistical analyses were performed on at least three independent biological replicates (n = 3). A p-value under 0.05 was considered statistically significant, and p-values are indicated in the figures with asterisks as follows; *p<0.05, **<p0.01, ***<0.001, and **** p<0.0001.

## IP, protein quantification, SDS-PAGE, and western blot analysis

IP of transfected 293T cell lysates was performed as described previously (*Schou et al., 2015*) using relevant antibody-conjugated beads (see Key resources table). Protein concentrations were measured using the *DC* Protein Assay Kit I from Bio-Rad (cat # 5000111) by following the manufacturer's protocol. For SDS-PAGE analysis of RPE1 and HSF cells, protein samples were prepared by lysis of cells with 95°C SDS-lysis buffer (1% SDS, 10 mM Tris-HCl, pH 7.4) and cell lysates transferred to Eppendorf tubes and heated shortly at 95°C. The samples were then sonicated two times for 30 s to shear DNA followed by centrifugation at 20,000 × *g* for 15 min at room temperature to pellet cell

debris. Supernatants were transferred to new Eppendorf tubes and an aliquot of each sample used for determination of protein concentration. Samples with equal concentrations of protein were prepared for SDS-PAGE analysis by addition of NuPAGE LDS Sample Buffer (4X) from Thermo Fisher Scientific (cat# NP0007) and 50 mM DTT. Samples from IP analysis in 293T cells, performed using modified EBC buffer (*Schou et al., 2015*), were prepared for SDS-PAGE in a similar fashion. Protein samples were heated at 95°C for 5 min before loading them on a Mini-PROTEAN TGX Precast Gel 4–15% from Bio-Rab Laboratories, Inc (cat# 456-1083 or cat #456-1086). PageRuler Plus Prestained Protein Ladder (Thermo Fisher Scientific, cat# 26619) was used as molecular mass marker. SDS-PAGE was performed using the Mini-PROTEAN Tetra System from Bio-Rad. Gels were run at 100 V for 15 min and 200 V for 45 min. Using the Trans-BLOT Turbo Transfer System from Bio-Rad Laboratories, Inc, the proteins were transferred at 1.3 A, 25 V for 10 min from the gel to a Trans-Blot Turbo Transfer Pack, Mini format 0.2 μm Nitrocellulose membrane (cat# 1704158). Ponceau-S solution was added to the membrane to visualize the proteins before blocking in 5% milk in Tris-buffered saline with Tween-20 (TBS-T; 10 mM Tris-HCl, pH 7.5, 150 mM NaCl, 0.1% Tween-20) for 2 hr at room temperature. Primary antibodies (see Key resources table) were diluted in 5% milk in TBS-T and incubated with the membrane overnight at 4°C. The membrane was washed three times 10 min at room temperature in TBS-T on a shaker before incubation with secondary antibody (see Key resources table), diluted in 5% milk in TBS-T, for 1 hr at room temperature. The membrane was washed three times 10 min at room temperature in TBS-T on a shaker. Finally, SuperSignal west Pico PLUS chemiluminescent Substrate (Thermo Scientific, cat# 34580) was mixed in a 1:1 ratio and added to the membrane for 5 min before development on a FUSION FX SPECTRA machine from Vilber.

## Quantitative MS and peptide identification

For SILAC-based MS, three distinct SILAC culture media were used: light- (Lys0 and Arg0); medium- (Lys4 and Arg6), and high-labeled media (Lys8 and Arg10). 293T cells were seeded in 100 mm Petri dishes and SILAC labeled for 1 week to ensure proper amino acid incorporation. After the incorporation phase, the cells were transfected with 2 μg of plasmid encoding FLAG-Ap80 (control), FLAG-CEP78, or FLAG-CEP78$^{L150S}$ using 6 μl FuGENE6, as described above, and then subjected to lysis using modified EBC buffer (*Douanne et al., 2019*) and FLAG IP as described above. Affinity-enriched proteins were digested with trypsin and the resulting peptides desalted prior to LCMS analysis using an Easy-nLC 1000 system (Thermo Scientific) connected to a Q Exactive HF-X mass spectrometer (Thermo Scientific). Peptides were separated by a 70 min gradient using increased buffer B (95% ACN, 0.5% acetic acid). The instrument was running in positive ion mode with MS resolution set at 60,000 for a scan range of 300–1700 m/z. Precursors were fragmented by higher-energy collisional dissociation (HCD) with normalized collisional energy of 28 eV. For protein identification and quantitation, the obtained MS raw files were processed by MaxQuant software version 1.6.1.0 (*Cox and Mann, 2008*) and searched against a FASTA file from UniProt.

## RNA-seq analysis

RNA was extracted using the Direct-zol RNA Miniprep Kit (Zymo Research) following the manufacturer's instructions. RNA degradation and contamination were monitored on 1% agarose gels. RNA purity was checked using the NanoPhotometer spectrophotometer (IMPLEN, CA, USA). RNA concentration was measured using Invitrogen Qubit RNA Assay Kit in Qubit 2.0 Fluorometer (Thermo Fisher Scientific). RNA integrity was assessed using the Agilent RNA Nano 6000 Assay Kit of the Bioanalyzer 2100 system (Agilent, CA, USA). A total amount of 3 μg RNA per sample was used as input material for the RNA sample preparations. Sequencing libraries were generated using NEBNext Ultra RNA Library Prep Kit for Illumina (New England BioLabs, USA) following the manufacturer's recommendations and index codes were added to attribute sequences to each sample. Briefly, mRNA was purified from total RNA using poly-T oligo-attached magnetic beads. Fragmentation was carried out using divalent cations under elevated temperature in NEBNext First Strand Synthesis Reaction Buffer (5X). First-strand cDNA was synthesized using random hexamer primer and M-MuLV Reverse Transcriptase (RNase H-). Second-strand cDNA synthesis was subsequently performed using DNA Polymerase I and RNase H. Remaining overhangs were converted into blunt ends via exonuclease/polymerase activities. After adenylation of 3′ ends of DNA fragments, NEBNext Adaptor with hairpin loop structure were ligated to prepare for hybridization. In order to select cDNA fragments

of preferentially 150–200 bp in length, the library fragments were purified with AMPure XP system (Beckman Coulter Life Sciences). Then 3 µl USER Enzyme (New England BioLabs) was used with size-selected, adaptor-ligated cDNA at 37°C for 15 min followed by 5 min at 95°C before PCR. Then PCR was performed with Phusion High-Fidelity DNA polymerase, Universal PCR primers and Index (X) Primer (New England BioLabs). At last, PCR products were purified (AMPure XP system) and library quality was assessed on the Agilent Bioanalyzer 2100 system. The clustering of the index-coded samples was performed on a cBot Cluster Generation System using HiSeq PE Cluster Kit cBot-HS (Illumina) according to the manufacturer's instructions. After cluster generation, the library preparations were sequenced on an Illumina Hiseq platform and 125 bp/150 bp paired-end reads were generated. Raw data (raw reads) of fastq format were firstly processed through in-house Perl scripts. In this step, clean data (clean reads) were obtained by removing reads containing adapter, reads containing poly-N and low-quality reads from raw data. At the same time, Q20, Q30, and GC content, the clean data were calculated. All the downstream analyses were based on the clean data with high quality. Reference genome and gene model annotation files were downloaded from genome website directly. Index of the reference genome was built using Bowtie v2.2.3 (*Langmead and Salzberg, 2012*), and paired-end clean reads were aligned to the reference genome using TopHat v2.0.12 (*Kim et al., 2013*). We selected TopHat as the mapping tool for that TopHat can generate a database of splice junctions based on the gene model annotation file and thus a better mapping result than other non-splice mapping tools. HTSeq v0.6.1 (*Anders et al., 2015*) was used to count the reads numbers mapped to each gene and then fragments per kilobase of transcript sequence per millions base pairs sequenced (FPKM) of each gene was calculated based on the length of the gene and reads count mapped to this gene. FPKM considers the effect of sequencing depth and gene length for the reads count at the same time, and is currently the most commonly used method for estimating gene expression levels (*Trapnell et al., 2010*). Prior to differential gene expression analysis, for each sequenced library, the read counts were adjusted by edgeR program package through one scaling normalized factor (*Robinson et al., 2010*). Differential expression analysis of two conditions was performed using the DEGSeq R package (1.20.0) (*Love et al., 2014*).

## Acknowledgements

We thank Søren Lek Johansen and Maria Schrøder Holm for expert technical assistance, Benedicte Schultz Kappel and Frida Roikjer Rasmussen for help with plasmid generation, and Nynne Christensen, Center for Advanced Bioimaging (CAB) Denmark, for help with 3D SIM acquisition. Dr. Kay Schink kindly provided plasmids for lentivirus particle generation and assisted in 3D SIM acquisition at OUH Radiumhospitalet. We are grateful to Drs. Peter K Jackson, Tomoharu Kanie, Francesc Garcia-Gonzalo, Laurence Pelletier, Brian David Dynlacht, Anne-Marie Tassin, Maria Gavilan, Rosa M Rios, Éric A Cohen, and Peter ten Dijke for reagents, and Dr. Michel Bornens for sharing original EM micrographs. This study was supported by grants from Independent Research Fund Denmark (#8020-00162B to PF and LBP, and #8021-00425A to JSA), from the Carlsberg Foundation (#CF18-0294 to LBP) and from the Swiss National Science Foundation (#176097 to CR).

## Additional information

### Funding

| Funder | Grant reference number | Author |
|---|---|---|
| Independent Research Fund Denmark | 8020-00162B | Pietro Farinelli<br>Lotte Bang Pedersen |
| Carlsberg Foundation | CF18-0294 | Lotte Bang Pedersen |
| Independent Research Fund Denmark | 8021-00425A | Jens S Andersen |
| Swiss National Science Foundation | 176097 | Carlo Rivolta |

The funders had no role in study design, data collection and interpretation, or the decision to submit the work for publication.

### Author contributions
André Brás Gonçalves, Conceptualization, Formal analysis, Supervision, Investigation, Visualization, Writing - original draft, Writing - review and editing; Sarah Kirstine Hasselbalch, Formal analysis, Investigation, Methodology, Writing - original draft; Beinta Biskopstø Joensen, Conceptualization, Formal analysis, Supervision, Investigation, Visualization, Methodology, Writing - original draft, Writing - review and editing; Sebastian Patzke, Formal analysis, Visualization, Writing - review and editing; Pernille Martens, Signe Krogh Ohlsen, Formal analysis, Investigation, Methodology; Mathieu Quinodoz, Konstantinos Nikopoulos, Formal analysis, Investigation, Visualization, Methodology; Reem Suleiman, Investigation; Magnus Per Damsø Jeppesen, Catja Weiss, Investigation, Methodology; Søren Tvorup Christensen, Investigation, Methodology, Writing - review and editing; Carlo Rivolta, Jens S Andersen, Formal analysis, Supervision, Funding acquisition, Visualization, Writing - original draft, Project administration, Writing - review and editing; Pietro Farinelli, Conceptualization, Formal analysis, Supervision, Funding acquisition, Investigation, Visualization, Methodology, Writing - original draft, Writing - review and editing; Lotte Bang Pedersen, Conceptualization, Supervision, Funding acquisition, Visualization, Writing - original draft, Project administration, Writing - review and editing

### Author ORCIDs
Søren Tvorup Christensen (ID) http://orcid.org/0000-0001-5004-304X
Lotte Bang Pedersen (ID) https://orcid.org/0000-0002-9749-3758

### Decision letter and Author response
Decision letter https://doi.org/10.7554/eLife.63731.sa1
Author response https://doi.org/10.7554/eLife.63731.sa2

## Additional files

### Supplementary files
• Transparent reporting form

### Data availability
All data generated or analysed during this study are included in the manuscript and supporting files. Source data files have been provided for Figure 3A and Figure 7-figure supplement 2.

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

# Appendix 1

**Appendix 1—key resources table**

| Reagent type (species) or resource | Designation | Source or reference | Identifiers | Additional information |
|---|---|---|---|---|
| Cell line (*Homo sapiens*) | hTERT-RPE1 | ATCC | Cat# CRL-4000 | Derived from human retinal pigment epithelium |
| Cell line (*H. sapiens*) | Puromycin-sensitive hTERT-RPE1 | Francesc Garcia-Gonzalo, Autonomous University of Madrid, Madrid, Spain | N/A | |
| Cell line (*H. sapiens*) | hTERT-RPE1 *CEP78* KO | *Ascari et al., 2020* | Clone #73 | |
| Cell line (*H. sapiens*) | hTERT-RPE1 *CEP78* KO | This study | Clone #44 | Generated as described in *Ascari et al., 2020*; contains px459-Cas9 plasmid inserted in exon 1 |
| Cell line (*H. sapiens*) | hTERT-RPE1 *CEP78* KO | This study | Clone #52 | Generated as described in *Ascari et al., 2020*; contains homozygous frameshift mutation in exon 1 of *CEP78* (single-nucleotide insertion in codon 10; GAC>GAAC) |
| Cell line (*H. sapiens*) | hTERT-RPE1 *CEP78* KO | This study | Clone #2 | Generated as described in *Ascari et al., 2020*; homozygous frameshift mutation in exon 1 of *CEP78* (single-nucleotide insertion in codon 10; GAC>GAAC) |
| Cell line (*H. sapiens*) | hTERT-RPE1 *CEP78* KO/ mNG-CEP78 | This study | N/A | Parental line is *CEP78* KO clone #73 |
| Cell line (*H. sapiens*) | hTERT-RPE1 *CEP78* KO/ mNG-CEP78$^{L150S}$ | This study | N/A | Parental line is *CEP78* KO clone #73 |
| Cell line (*H. sapiens*) | hTERT-RPE1 WT/ mNG-CEP78 | This study | N/A | Parental line is puromycin-sensitive hTERT-RPE listed above |
| Cell line (*H. sapiens*) | hTERT-RPE1-BFP-Cas9/ pMCB306 (pool) | *Kanie et al., 2017* | N/A | Control cell line for *CEP350* and *FOP* KO lines |
| Cell line (*H. sapiens*) | hTERT-RPE1-BFP-Cas9/ *CEP350* KO (pool) | *Kanie et al., 2017* | N/A | |
| Cell line (*H. sapiens*) | hTERT-RPE1-BFP-Cas9/ *FOP* KO (pool) | *Kanie et al., 2017* | N/A | |
| Cell line (*H. sapiens*) | 293T | ATCC | Cat# CRL-3216 | Derived from human embryonic kidney |
| Cell line (*H. sapiens*) | Control skin fibroblasts | *Nikopoulos et al., 2016* | N/A | Primary cells |

*Continued on next page*

*Appendix 1—key resources table continued*

| Reagent type (species) or resource | Designation | Source or reference | Identifiers | Additional information |
|---|---|---|---|---|
| Cell line (*H. sapiens*) | Compound heterozygous skin fibroblasts with two *CEP78* mutations: c.449T>C; p. Leu150Ser and c.1462–1G>T | *Ascari et al., 2020* | F3: II:1 | Primary cells; these cells do not express detectable levels of CEP78 *Ascari et al., 2020* |
| Cell line (*H. sapiens*) | Compound heterozygous skin fibroblasts with two *CEP78* mutations: c.633delC; p.Trp212Glyfs*18 and c.499+5G>A; IVS + 5G>A | *Nikopoulos et al., 2016* | 2702 r34, individual II-3 | Primary cells; these cells do not express detectable levels of CEP78 *Nikopoulos et al., 2016* |
| Cell line (*H. sapiens*) | Homozygous substitution in the first invariant base of intron three splice donor site in *CEP78*: c.499+1G>T (IVS3 +1G>T) | *Nikopoulos et al., 2016* | KN10, individual II-1 | Primary cells; these cells express almost undetectable levels of CEP78 *Nikopoulos et al., 2016* |
| Strain, strain background (*Escherichia coli*) | DH10B | Lab stock | N/A | |
| Antibody | Anti-acetylated-tubulin (mouse monoclonal) | Sigma-Aldrich | Cat# T7451 | IFM (1:2000) |
| Antibody | Anti-α-tubulin (mouse monoclonal) | Sigma-Aldrich | Cat# T5168 | WB (1:10,000) |
| Antibody | Anti-ARL13B (rabbit polyclonal) | Proteintech | Cat# 17711-1-AP | IFM (1:500) |
| Antibody | Anti-CEP78 (rabbit polyclonal) | Bethyl Laboratories | Cat# A301-799A-M | IFM (1:200) WB (1:500) |
| Antibody | Anti-CEP164 (rabbit polyclonal) | Sigma-Aldrich | Cat# HPA037606 | IFM (1:500) |
| Antibody | Anti-CEP350 (mouse monoclonal) | Abcam | Cat# Ab219831 (CL3423) | IFM (1:500) |
| Antibody | Anti-CEP350 (rabbit polyclonal) | Novus Biologicals | Cat# NB100-59811 | IFM (1:200) |
| Antibody | Anti-CP110 (rabbit polyclonal) | Proteintech | Cat# 12780-1-AP | IFM (1:200) WB (1:1500) |
| Antibody | Anti-DCTN1/P-150 (mouse monoclonal) | BD Biosciences | Cat# 610474 | IFM (1:500) |
| Antibody | Anti-EDD1 (rabbit polyclonal) | Bethyl Laboratories | Cat# A300-573A-M | IFM (1:100) |

*Continued on next page*

*Appendix 1—key resources table continued*

| Reagent type (species) or resource | Designation | Source or reference | Identifiers | Additional information |
|---|---|---|---|---|
| Antibody | Anti-FLAG (mouse monoclonal) | Sigma-Aldrich | Cat# F1804 | IFM (1:500) WB (1:1000) |
| Antibody | Anti-FLAG (rabbit polyclonal) | Sigma-Aldrich | Cat# F7425 | WB (1:1000) |
| Antibody | Anti-FGFR1OP/ FOP (mouse monoclonal) | Novus Biologicals | Cat# H00011116-M01 | WB (1:500) |
| Antibody | Anti-GAPDH (rabbit polyclonal) | Cell Signaling Technology | Cat# 2118 | WB (1:1000) |
| Antibody | Anti-GFP (rabbit polyclonal) | Sigma-Aldrich | Cat# SAB4301138 | WB (1:500) |
| Antibody | Anti-GFP (rabbit polyclonal) | Santa Cruz Biotechnologies, Inc | Cat# sc-8834 | WB (1:500) |
| Antibody | Anti-Myc (mouse monoclonal, 9B11) | Cell Signaling Technology | Cat# 2276S | WB (1:1000) |
| Antibody | Anti-phospho Rb (S807/811) (mouse monoclonal) | Cell Signaling Technology | Cat# 9308 | WB (1:500) |
| Antibody | Anti-VPRBP (rabbit polyclonal) | Bethyl Laboratories | Cat# A301-877A-M | WB (1:200) |
| Antibody | Polyclonal Goat Anti-Mouse Immunoglobulins/ Horseradish Peroxidase conjugated | Dako | Cat# P0447 | WB (1:10,000) |
| Antibody | Polyclonal Swine Anti-Rabbit Immunoglobulins/ Horseradish Peroxidase conjugated | Dako | Cat# P0399 | WB (1:10,000) |
| Antibody | Donkey anti-Mouse IgG (H+L) Highly Cross-Adsorbed Secondary Antibody/ Alexa Fluor 488-conjugated | Thermo Fisher Scientific | Cat# A21202 | IFM (1:500) |
| Antibody | Donkey anti-Mouse IgG (H+L) Highly Cross-Adsorbed Secondary Antibody/ Alexa Fluor 568-conjugated | Thermo Fisher Scientific | Cat # A10037 | IFM (1:500) |

*Continued on next page*

*Appendix 1—key resources table continued*

| Reagent type (species) or resource | Designation | Source or reference | Identifiers | Additional information |
|---|---|---|---|---|
| Antibody | Donkey anti-Rabbit IgG (H+L) Highly Cross-Adsorbed Secondary Antibody/Alexa Fluor 488-conjugated | Thermo Fisher Scientific | Cat #A21206 | IFM (1:500) |
| Antibody | Donkey anti-Rabbit IgG (H+L) Highly Cross-Adsorbed Secondary Antibody/Alexa Fluor 568-conjugated | Thermo Fisher Scientific | Cat# A10042 | IFM (1:500) |
| Antibody | AffiniPure Donkey Anti-Mouse IgG (H+L)/Alexa Fluor 647-conjugated | Jackson ImmunoResearch | Cat# 715-605-150 | IFM (1:1000) |
| Antibody | Donkey Anti-Rabbit IgG H and L Secondary Antibodies/Dylight 550-conjugated | Abcam | Cat# ab96892 | IFM (1:1000) |
| Enzyme | USER Enzyme | New England BioLabs | Cat# M5505L | |
| Enzyme | Phusion High-Fidelity DNA polymerase | New England BioLabs | Cat# M0530L | |
| Recombinant DNA reagent | *H. sapiens* Angiomotin p80 (Ap80)/pCMV-FLAG | *Yi et al., 2013* | N/A | |
| Recombinant DNA reagent | *H. sapiens* CEP78 aa 2–217/pEGFP-C1 | This study | CEP78 N-terminus (N) | PCR-amplified using primers h.CEP78.4f.kpn1 and hsEGFP-CEP78_Rv (651) |
| Recombinant DNA reagent | *H. sapiens* CEP78 aa 203–403/pEGFP-C1 | This study | CEP78 middle (M) | PCR-amplified using primers hsEGFP-CEP78_Fwd (607) and hsEGFP-CEP78_Rv (1209) |
| Recombinant DNA reagent | *H. sapiens* CEP78 aa 2–403/pEGFP-C1 | This study | CEP78 N-terminus plus middle (NM) | PCR-amplified using primers h.CEP78.4f.kpn1 and hsEGFP-CEP78_Rv (1209) |
| Recombinant DNA reagent | *H. sapiens* CEP78 aa 395–722/pEGFP-C1 | This study | CEP78 C-terminus (C) | PCR-amplified using primers hsEGFP-CEP78_Fwd (1186) and hCEP78.rv_stop.bamh1 |
| Recombinant DNA reagent | *H. sapiens* CEP78/pEGFP-C1 | This study | N/A | PCR-amplified using primers h.CEP78.4f.kpn1 and hCEP78.rv_stop.bamh1 |
| Recombinant DNA reagent | *H. sapiens* CEP78/p3xFLAG-CMV/DEST | This study | p3xFLAG-CEP78 | |

*Continued on next page*

*Appendix 1—key resources table continued*

| Reagent type (species) or resource | Designation | Source or reference | Identifiers | Additional information |
|---|---|---|---|---|
| Recombinant DNA reagent | *H. sapiens* CEP78[L150S]/ p p3xFLAG-CMV/ DEST | This study | p3xFLAG-CEP78[L150S] | |
| Recombinant DNA reagent | Plasmid pEGFP-C1 (empty vector) | TaKaRa Bio | Cat# 6084-1 | |
| Recombinant DNA reagent | Gateway pDONR201 | Invitrogen/Thermo Fisher Scientific | Cat# 11798-014 | |
| Recombinant DNA reagent | p3xFLAG-CMV-10 | Sigma-Aldrich | Cat# E7658 | Cloning produces in-frame 3xFLAG tag in the N-terminus |
| Recombinant DNA reagent | p3xFLAG-CMV/ DEST | Gift from Ronald Roepman lab, Radboud University Medical Center, Nijmegen, NL | pDEST306 | p3xFLAG-CMV-10 with Gateway Rf-B cassette inserted into the blunted HindIII/BamHI site |
| Recombinant DNA reagent | *H. sapiens* CEP350 aa 1–983/pCS2+ with a 6xmyc tag | *Eguether et al., 2014*; *Hoppeler-Lebel et al., 2007* | CAP N | |
| Recombinant DNA reagent | *H. sapiens* CEP350 aa 2990–3117/pCS2+ with a 6xmyc tag | *Eguether et al., 2014*; *Hoppeler-Lebel et al., 2007* | CAP C | |
| Recombinant DNA reagent | *H. sapiens* VPRBP/ pCMV-Myc | *Belzile et al., 2007* | Myc-VPRBP | Cloned in Sal1 and Not1 sites |
| Sequence-based reagent | *H. sapiens CEP78* ORF/ GeneCopoiea pShuttleGateway PLUS ORF | GeneCopoiea | NCBI reference Sequence NM_001098802.3 | |
| Sequence-based reagent | *H. sapiens CEP78* PCR primer | Sigma-Aldrich | attB forward | 5′-GGGACAAGTTGTACAAAAAACAG GCTTCATCGACTCCGTGAAGCTGC-3′ |
| Sequence-based reagent | *H. sapiens CEP78* PCR primer | Sigma-Aldrich | attB reverse | 5-GGGGACCACTTTGTACAAGAAAGCTGG GTTTCAGGAATGCAGGTCCTTTCCAG-3′ |
| Sequence-based reagent | *H. sapiens CEP78* PCR primer | Sigma-Aldrich | CEP78 mut_F | 5′-agagacaggtgcaccgaagaagccgatttattcaatc-3′ |
| Sequence-based reagent | *H. sapiens CEP78* PCR primer | Sigma-Aldrich | CEP78 mut_R | 5′-gattgaataaatcggcttcttcggtgcacctgtctct-3′ |
| Sequence-based reagent | *H. sapiens CEP78* PCR primer | Eurofins Genomics | hCEP78.4f.kpn1 | 5′-AAGGTACC ATCGACTCCGTGAAGCTGCG-3′ |
| Sequence-based reagent | *H. sapiens CEP78* PCR primer | Eurofins Genomics | hsEGFP-CEP78_Rv (651) | 5′-AAGGATCCTTAGCGA AGACTCTCAGCCCAG-3′ |
| Sequence-based reagent | *H. sapiens CEP78* PCR primer | Eurofins Genomics | hsEGFP-CEP78_Fwd (607) | 5′- GGGGTACC CAGACCATGAGAAGGCATGA -3′ |

*Continued on next page*

*Appendix 1—key resources table continued*

| Reagent type (species) or resource | Designation | Source or reference | Identifiers | Additional information |
|---|---|---|---|---|
| Sequence-based reagent | *H. sapiens CEP78* PCR primer | Eurofins Genomics | hsEGFP-CEP78_Rv (1209) | 5'-GGGGATCC TTAACCCCTGTGTCTTTTTGCAC-3' |
| Sequence-based reagent | *H. sapiens CEP78* PCR primer | Eurofins Genomics | hsEGFP-CEP78_Fwd (1186) | 5'-GGGGTACC GCAGAACGTGCAAAAAGACA-3' |
| Sequence-based reagent | *H. sapiens CEP78* PCR primer | Eurofins Genomics | hCEP78.rv_stop. bamh1 | 5'-GAGGATCC ACAGGAATGCAGGTCCTTTC-3' |
| Sequence-based reagent | Control siRNA | Eurofins Genomics | N/A | 5'-UAAUGUAUUGGAAGGCA-3' |
| Sequence-based reagent | *H. sapiens CP110* siRNA | Eurofins Genomics / *Spektor et al., 2007* | N/A | 5'-GCAAAACCAGAAUACGAGATT-3' |
| Chemical compound, drug | DAPI (4', 6-diamidino-2-phenylindole) | Thermo Fisher Scientific | Cat# D1306 | Stored at −20°C as stock solution of 20 µg/ml in $H_2O$ |
| Chemical compound drug | FuGENE6 | Promega | Cat# E2692 | |
| Chemical compound, drug | DharmaFECT Duo Transfection reagent | Dharmacon | Cat# T-2010-03 | |
| Software, algorithm | cellSens Dimension | Olympus | Version 1.18 | |
| Software, algorithm | Zen Black 2012 | Zeiss | Version 2012 | |
| Software, algorithm | Adobe Photoshop | Adobe | Version 21.0.1 | |
| Software, algorithm | Adobe Illustrator | Adobe | Version 24.2.1 | |
| Software, algorithm | ImageJ | NIH | Mac OS X or Windows | |
| Software, algorithm | GraphPad Prism 6.0 | GraphPad Software Inc | Mac OS X or Windows | |
| Software, algorithm | MaxQuant | *Cox and Mann, 2008* | Version 1.6.1.0 | |
| Other | Anti-FLAG M2 Affinity Gel | Sigma-Aldrich | Cat# A2220 | |
| Other | Anti-c-Myc Agarose Affinity Gel antibody produced in rabbit | Sigma-Aldrich | Cat# A7470-1 ml | |
| Other | GFP-Trap Agarose | ChromoTek GmbH | Cat# gta-20 | |
| Other | QuickChange Lightning Site-Directed Mutagenesis Kit | Agilent | Cat# 210518 | |
| Other | Direct-zol RNA Miniprep Kit | Zymo Research | Cat# R2051 | |

*Continued on next page*

*Appendix 1—key resources table continued*

| Reagent type (species) or resource | Designation | Source or reference | Identifiers | Additional information |
|---|---|---|---|---|
| Other | Agilent RNA Nano 6000 Assay Kit | Agilent | Cat# 5067-1511 | |
| Other | NEBNext Ultra RNA Library Prep Kit for Illumina | New England BioLabs | Cat# E7530L | |
| Other | Invitrogen Qubit RNA Assay Kit | Thermo Fisher Scientific | Cat# 32852 | |
| Other | AMPure XP system | Beckman Coulter Life Sciences | Cat# A63881 | |

