## [Decision Letter]

**Acceptance summary:**

In this paper Goncalves et al. reveal how loss-of-function of the centriolar protein CEP78 is linked to the ciliopathy cone-rod dystrophy with hearing loss (CRDHL). Using KO cell lines, patient cells, and proteomics, the authors show that CEP78 interacts with CEP350 for centriole localization and with the EDD1 E3 ubiquitin ligase for mediating degradation of CP110, a negative regulator of ciliogenesis. CEP78 loss-of-function results in stabilization of CP110, suppressing ciliogenesis.

**Decision letter after peer review:**

Thank you for submitting your article "CEP78 functions downstream of CEP350 to control biogenesis of primary cilia by negatively regulating CP110 levels" for consideration by *eLife*. Your article has been reviewed by 3 peer reviewers, one of whom is a member of our Board of Reviewing Editors, and the evaluation has been overseen by Piali Sengupta as the Senior Editor. The reviewers have opted to remain anonymous.

The reviewers have discussed the reviews with one another and the Reviewing Editor has drafted this decision to help you prepare a revised submission.

Summary:

In this paper Goncalves et al. investigated the role of the centriolar protein CEP78 in formation and growth of primary cilia. CEP78 is mutated in patients suffering from cone-rod dystrophy with hearing loss (CRDHL), a ciliopathy. The current paper aims to clarify how CEP78 affects cilia and what the molecular basis of its action is. Contrary to previous findings, the authors, using knockout and CRDHL patient cells, identify CEP78 as a negative regulator of CP110 that promotes cilia assembly and limits cilium length. They also identify CEP350 as factor acting upstream in this pathway by promoting CEP78 centrosome recruitment.

The reviewers agree that overall the data is of good quality and that the manuscript is timely. The paper contains novel and important findings regarding the role of CEP78 in regulating cilium assembly and length.

However, they also felt that settling the conflict with previously published data and advancing our understanding of the cellular roles of CEP78 would require a more in-depth analysis than what is currently presented.

Essential revisions:

1. Rescue Experiments:

a) The authors need to establish proper rescue with CEP78 WT and mutant (Figure 2).

Basically, there is no rescue of the phenotype here, which would normally be interpreted to mean that the phenotype is not specific to CEP78 loss. If the authors think this is related to expression levels, they should demonstrate this. Appropriate levels should rescue the phenotype. This could be done e.g. by (i) expression under a weaker promoter, (ii) construction of cell lines where WT or L150S are stably expressed with suitable levels or expression can be induced and modulated.

b) The authors need to establish proper rescue by CP110 siRNA (Figure 8). Currently there does not seem to be a statistically significant difference between CEP78-KO+siMock and CEP78-KO+siCP110 conditions. Rather than only looking at total levels, the authors should also compare CP110 levels at mother centrioles at the onset of ciliogenesis, which is what ultimately matters for ciliogenesis.

2. The authors should test to what extent WT and L150S rescue total and centrosomal CP110 levels in CEP78-KO cells, which would support their model.

3. Proteasomal regulation of CP110 levels:

The authors seem to imply that CEP78 regulates centrosomal recruitment of EDD1, which promotes CP110 degradation. To test if CP110 half-life is increased in CEP78-KO cells, cycloheximide chase experiments should be performed. Together with the mRNA data in Figure 7-suppl.2, this would rule out that increased CP110 protein in KO cells is due to increased transcription or translation.

[Editors' note: further revisions were suggested prior to acceptance, as described below.]

Thank you for resubmitting your work entitled "CEP78 functions downstream of CEP350 to control biogenesis of primary cilia by negatively regulating CP110 levels" for further consideration by *eLife*. Your revised article has been evaluated by Piali Sengupta as the Senior Editor, and a Reviewing Editor.

All three reviewers appreciate the addition of new data including rescue experiments to the manuscript. Overall the revisions have addressed most of the concerns and have improved the manuscript. Prior to acceptance we would like to ask the authors to address the following remaining issues.

The authors present and discuss the rescue data in Figure 2 as "partial rescue", yet no statistical testing was done on the actual rescue pair: CEP78 KO and CEP78 KO + mNG-CEP78. This should be added. To make it complete, comparison between CEP78 KO and CEP78 KO + mNG-CEP78 L150S would also be useful.

Are the observed differences between CEP78 KO and CEP78 KO + mNG-CEP78 indeed statistically significant to conclude partial rescue?

Depending on the outcome the authors could repeat the experiment to improve the statistics or present the data differently in the results. In addition, the rescue data should be discussed accordingly in the Discussion section.

---

## [Author Response]

Essential revisions:1. Rescue Experiments:a) The authors need to establish proper rescue with CEP78 WT and mutant (Figure 2).Basically, there is no rescue of the phenotype here, which would normally be interpreted to mean that the phenotype is not specific to CEP78 loss. If the authors think this is related to expression levels, they should demonstrate this. Appropriate levels should rescue the phenotype. This could be done e.g. by (i) expression under a weaker promoter, (ii) construction of cell lines where WT or L150S are stably expressed with suitable levels or expression can be induced and modulated.

We have generated stable cell lines expressing low levels of mNG-tagged versions of CEP78 or CEP78^L150S^, and we have analyzed cilia length and frequency in these lines. Our results, now included in revised Figure 2C, D and Figure 2—figure supplement 3, show that mNG-CEP78 can at least partially rescue the cilia frequency and length phenotype of the *CEP78* KO cells while mNG-CEP78 ^L150S^ cannot. We have updated the text accordingly.

b) The authors need to establish proper rescue by CP110 siRNA (Figure 8). Currently there does not seem to be a statistically significant difference between CEP78-KO+siMock and CEP78-KO+siCP110 conditions. Rather than only looking at total levels, the authors should also compare CP110 levels at mother centrioles at the onset of ciliogenesis, which is what ultimately matters for ciliogenesis.

We have performed additional repeats of the CP110 siRNA KD experiments in wild type and CEP78 KO cells, which indicate statistically significant differences in cilia frequency between CEP78-KO+siMock and CEP78-KO+siCP110 conditions (new Figure 8D). However, the elongated length phenotype in the CEP78 KO cells was not rescued by CP110 depletion (new Figure 8E). This indicates that CEP78 affects ciliogenesis and ciliary length by distinct mechanisms, which is also in agreement with our new data using a truncated CEP78 KO clone (clone #44); this clone has fewer cilia than WT control cells, but the length of remaining cilia is normal (new Figure 1—figure supplement 1). We have also compared CP110 levels at centrosomes during onset of ciliogenesis, and under relevant conditions, as requested. The new data is shown in Figure 8—figure supplement 1, and confirms that CP110 levels at the centrosome are indeed reduced in CP110 siRNA-treated cells as expected. Manuscript text has been updated accordingly.

2. The authors should test to what extent WT and L150S rescue total and centrosomal CP110 levels in CEP78-KO cells, which would support their model.

We have performed western blot analysis of our new WT and CEP78 KO lines stably expressing mNG-CEP78 or mNG-CEP78^L150S^ and find that CP110 levels are similar in these lines (new Figure 7—figure supplement 1C). That mNG-CEP78^L150S^ could rescue cellular CP110 levels was a bit surprising to us since it does not localize to the centrosome, but given that the mNG fusion protein is expressed at ca. 3-4 times the level of endogenous CEP78 and retains some capacity to bind VPR-BP, we think that it might bind and activate some VPR-BP in the cytosol and thereby negatively regulate overall cytosolic levels of CP110, but without affecting local CP110 levels at the centrosome. Due to limitations of our microscope (weak signal in blue channel) it was not possible to reliably quantify mother and daughter centriole levels of CP110 (in red) in the mNG-CEP78 (in green) expressing lines when using a blue cilia/centrosome marker. Please note that our model is supported by several additional experiments that we performed for the revised manuscript (see above).

3. Proteasomal regulation of CP110 levels:The authors seem to imply that CEP78 regulates centrosomal recruitment of EDD1, which promotes CP110 degradation. To test if CP110 half-life is increased in CEP78-KO cells, cycloheximide chase experiments should be performed. Together with the mRNA data in Figure 7-suppl.2, this would rule out that increased CP110 protein in KO cells is due to increased transcription or translation.

We performed the requested experiments, which are included in the Figure 7—figure supplement 4 in the revised manuscript.

[Editors' note: further revisions were suggested prior to acceptance, as described below.]

All three reviewers appreciate the addition of new data including rescue experiments to the manuscript. Overall the revisions have addressed most of the concerns and have improved the manuscript. Prior to acceptance we would like to ask the authors to address the following remaining issues.The authors present and discuss the rescue data in Figure 2 as "partial rescue", yet no statistical testing was done on the actual rescue pair: CEP78 KO and CEP78 KO + mNG-CEP78. This should be added. To make it complete, comparison between CEP78 KO and CEP78 KO + mNG-CEP78 L150S would also be useful.Are the observed differences between CEP78 KO and CEP78 KO + mNG-CEP78 indeed statistically significant to conclude partial rescue?Depending on the outcome the authors could repeat the experiment to improve the statistics or present the data differently in the results. In addition, the rescue data should be discussed accordingly in the Discussion section.

We are thankful for this comment and apologize for the oversight. We also realized we had included the “wrong” datasets for WT and *CEP78* KO in the original Figures 2C, D. The original datasets were from Figure 1B, C, but these datasets were not acquired in parallel with those for the rescue lines. So, we replaced the original WT and *CEP78* KO datasets in Figure 2C, D with new datasets that we acquired in parallel with datasets for the rescue lines (acquired in conjunction with Figure 1—figure supplement 1), and we redid the statistics for Figure 2C, D. We now observe full rescue of the ciliation frequency phenotype with mNG-CEP78 but not with mNG-CEP78^L150S^. We also find that mNG-CEP78 can promote ciliary shortening of the KO cells, and while mNG-CEP78^L150S^ also promotes ciliary shortening in these cells, it does so less efficiently than mNG-CEP78 wild type. We have replaced Figure 2C, D with the new data and we have updated the figure legend, results and Discussion sections accordingly.